# Does Your Video-language Model Actually Understand the Language Input?

## Abstract

Driven by the wave of Large Language Models (LLMs), Video-Language Models (VLMs) have become a significant yet challenging technology to bridge the gap between video and text. Although previous VLM works have made significant progress, almost all of them implicitly assume that all the texts are predefined by the specific template. In real-world applications, such an assumption is impossible to satisfy, since predefining all the texts is extremely time-consuming and labor-intensive. Besides, these predefined text inputs are too strict and user-unfriendly, limiting their applications. It is observed that given a video input, texts with similar semantics lead to various performances. To this end, in this paper, we propose a novel text-augmented VLM method to improve video-text fusion by text rewriting. Specifically, we first generate various text samples from the original ones based on the pre-trained LLM to target specific text components. A multi-level contrastive learning module is designed to mine the coarse-grained language information. Moreover, we also propose an attribute-based text reasoning strategy to learn fine-grained textual semantics. Extensive experiments on many video-language tasks show that the proposed method can serve as the plug-and-play module to effectively improve the performance of state-of-the-art VLM works.

## 1 Introduction

Due to remarkable success, Large Vision-Language Models (LVLMs) have attracted more and more attention (Tian et al., 2024; Fan et al., 2024; Kim et al., 2024). LVLMs require cooperation from both computer vision and natural language processing for precise semantic alignment and have a wide range of applications such as video summarization and video question answering. Benefiting from the strong knowledge integration ability in large language models (LLMs), LVLMs show superior performances in solving complex image-language tasks by utilizing appropriate human-instructed prompts (Hakim et al., 2023; Duan et al., 2024; Jung et al., 2024). Since the real-world videos contain much temporal information, LVLMs still have difficulty to handle the real-world videos. Besides, the sentence text is the most important input that accompanies the video due to its human-friendly and descriptive nature.

Current video-language models contain three main popular yet challenging tasks: video question answering (VideoQA) (Gao et al., 2023; Yu et al., 2024), video sentence grounding (VSG) (Zhang et al., 2023b; Qi et al., 2024) and video-text retrieval (VTR) (Zhu et al., 2023; Zhang et al., 2023a). Video QA is a significant multi-modal task where a model is given a video along with a natural language question about the video content, and it must generate or select the correct answer. The task requires the model to understand the visual cues in the video, as well as the language of the question, to provide relevant and accurate responses. Given a language text and an untrimmed video, VSG aims at retrieving the start and end timestamps of the target video moment, semantically according to a sentence text. Given a language text, VTR targets to retrieve relevant videos from a large video database, which can be either text-based (text-to-video retrieval) or video-based (video-to-video retrieval). The significant goal of VTR is to find videos that best match the given input by analyzing visual content, actions, and sometimes audio cues. Their performance in these downstream tasks depends on their capability to extract video features and align them with text features. Some methods only perform data augmentations on the video input to improve the model robustness during training in every epoch. In contrast, existing methods only utilize predefined texts without any augmentation. In real-world applications, these sentence texts with similar textual semantics might be inputted

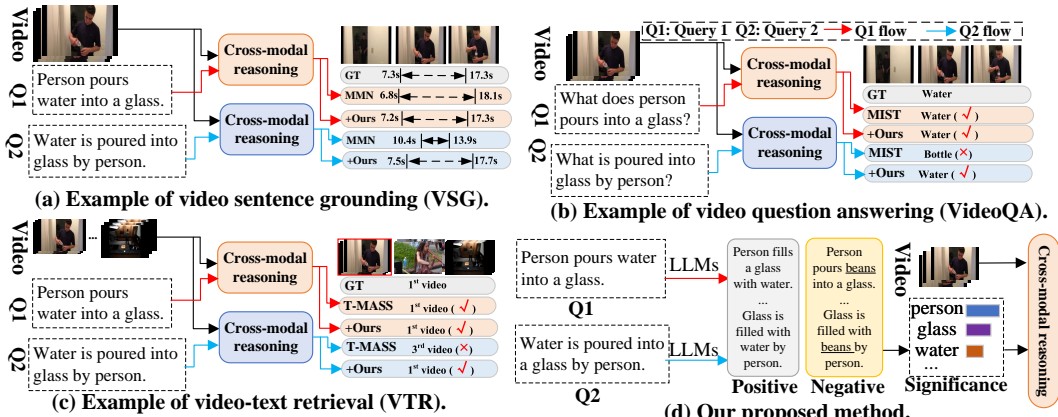

Figure 1: (a-c) Example of the VLM tasks (VSG, VideoQA and VTR), where our method can serve as a plug-and-play module for previous VLM models to enhance their efficiency. (d) Pipeline of our proposed method.

with different structure/vocabulary variations from various users. As shown in Figure 1(b), the text ("Person pours water into a glass") shares the same semantics as the text ("Water is poured into a glass by person"). However, previous methods yield dissimilar grounding results in Figure 1. The main reason is that these methods cannot utilize their weak text encoder to learn discriminative textual representations, which illustrates the significance of handling the text variations. Therefore, it is important to ensure that the designed VLM-based model is robust enough to deal with various texts with different templates. However, existing language augmentation approaches are not sufficiently effective to integrate the multi-modal inputs. Some methods target to replace or mask some words in a sentence, which only brings limited influence in diversifying the text structure/vocabulary. It is comparatively weaker than video augmentations. The target language augmentation approach should effectively rewrite sentence texts while reserving the core textual semantics. The approach is urgently required for model training to achieve the best results.

In this paper, we propose a simple yet highly effective framework to improve the robustness and performance of VLMs. Specifically, we leverage the large language models (LLM) to generate multiple variants of each text in the video-text pairs. To obtain different text types, we generate a small set of variation-origin by two strategies: LLM-based datasets and existing original datasets. After obtaining the variation-origin pairs, we utilize them as examples to prompt LLM model for diversifying all the texts in the VSG datasets. Different from previous sentence augmentation works that only change some words to preserve sentence structures, LLM has a strong language processing ability to generate rich variations for diverse text inputs due to their extensive training datasets and emergent properties. Based on the above sentence augmentation, each video corresponds to diverse texts. Moreover, we introduce GPT (Radford, 2018) to generate various hard negative texts from the original (anchor) texts by changing different sentence parts. In particular, we utilize precise prompt engineering to modify specific parts of the sentence with the rest parts unchanged. Also, we generate positive samples that lie relatively far from the anchor in the embedding space. To further understand the latent textual semantics, we design an attribute-based text reasoning strategy for fine-grained text mining. To analyze the relative significance of each sentence part, we incorporate these generated samples by a weighted contrastive loss function. With these diverse texts, we target to train VLM models with augmentation from the text perspective. For convenience, we randomly choose a text augmentation from multiple diverse texts.

Our main contributions are summarized as follows: 1) We make the first attempt to explore the effect of the template-free text for the robust VLM task, where we localize the target activity by a user-friendly text with any form instead of a predefined text. Also, we propose a novel framework that utilizes the LLMs to generate positive and negative texts, where each negative text is used to highlight a sentence component. 2) To obtain diverse positive and negative texts, we augment the text by both word-level and structure-level for rewriting texts. To selectively integrate these generated texts, we design two modules (generating multi-level texts module and attribute-based text reasoning module) to understand the text input from different granularities. Besides, a weighted contrastive loss is introduced to integrate these sentence components by assigning adaptive weights to these components. 3) For three downstream tasks (video sentence grounding, video question an-

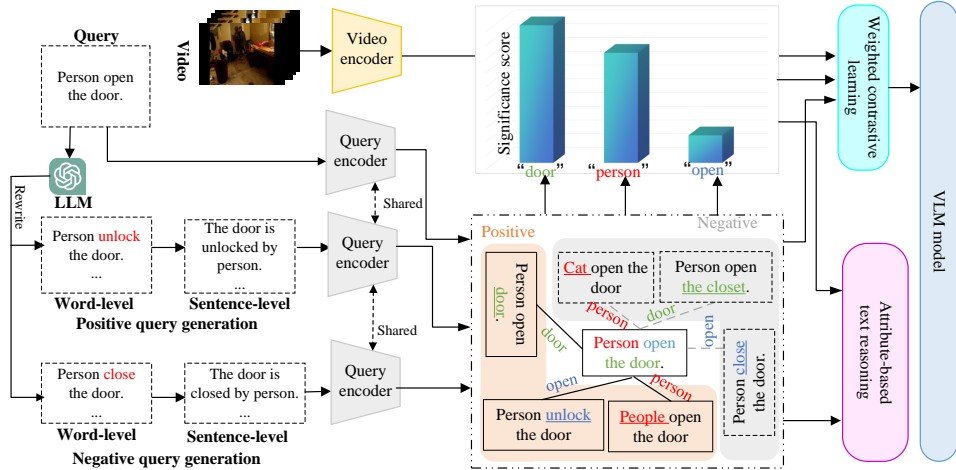

Figure 2: Illustration of our proposed framework. Best viewed in color.

swering and video-text retrieval), we conduct experiments on many popular yet challenging datasets. Extensive results show that our proposed model outperforms existing approaches by a large margin. Moreover, our method can serve as a plug-and-play module for state-of-the-art VLM methods.

## 2 RELATED WORKS

**Large vision-language models.** The breakthrough of LLMs in language-oriented tasks (Ma et al., 2024; Du et al., 2024) and the emergence of GPT-4 have prompted researchers to explore the potential of LLMs in assisting with a range of tasks across multi-modal scenarios (Carolan et al., 2024; Yin et al., 2024). This has led to the development of a new field, namely large vision-language models (LVLMs). A variety of strategies and models have been proposed to address the discrepancy between text and other modalities. Some works employ learnable texts to extract visual information and generate language using LLMs conditioned on the visual features. Models including GPT-4o, MiniGPT-4 and LLaVA learn simple projection layers to align the visual features from visual encoders with text embeddings for LLMs. Additionally, parameter-efficient fine-tuning is adopted by introducing lightweight trainable adapters into models. Several benchmarks have verified that LVLMs demonstrate satisfactory performance on visual perception and comprehension.

Although these methods have achieved promising results, all of them heavily rely on correctly aligned multi-modal datasets. Therefore, it is highly expected to develop a VLM model that is robust to different texts with similar semantics, which has not been studied as far as we know. Thus, *we make the first attempt to reveal the text understanding problem in VLM task and propose to eliminate the negative impact of the different texts with any template.* **More details in Section A.2.**

## 3 METHODOLOGY

We elaborate on the proposed method, which strengthens the text encoder to obtain consistent representations for various semantically similar texts in real-world multi-modal datasets (*e.g.*, Charades-STA (Sigurdsson et al., 2016)), multiple semantics-similar texts often share a video moment with the target activity. For example, "Person opens the door" and "The door is opened by person" have similar semantics. Since the text template is fixed, it is still challenging to diversify the text input. Thus, we design a text augmentation module to generate semantically similar texts. The overall framework is shown in Figure 2.

**Problem statement.** Due to the strong language processing ability of of LLMs, we utilize LLM to generate various texts by replacing different components for simulating the practical labeling process in the format-free setting. We denote $\{Q_1, \ldots, Q_M\}$ as the textual input set in the VLM task, where $M$ denotes the total number of sentences. Previous VLM methods (Yu et al., 2023; Wang et al., 2022b; 2024) cannot well handle these texts with similar semantics since they do not fully understand the textual information in the sentence. To address the existing models' limitations in correlating major sentence parts with suitable video representations, we present a novel plug-and-

play method for generating negative and positive samples targeting specific sentence parts. These samples facilitate improved perception of specific parts of the sentence, eventually enhancing the understanding of video-language correlation. We use the generated samples as auxiliary samples alongside the original training samples by employing a novel weighted contrastive loss. The proposed approach is application-agnostic and can be adopted successfully in the multi-modal task.

### 3.1 GENERATING MULTI-LEVEL TEXTS FOR COARSE-GRAINED LANGUAGE ALIGNMENT

By treating original texts as anchors, we target to leverage LLMs for generating positive and negative texts to fully understand the texts, where we regard the generated text sharing similar semantics as a positive text; otherwise, the generated text is negative. For the $m$-th text, we denote the generated positive text as $P_m$ and the negative text as $N_m$ . To obtain diverse generated texts, we adopt three text rewriting approaches: human-based rewriting, chat robot-based rewriting and open-source LLM-based rewriting in Section A.1. For convenience, we term the pair of generated text and original text as variation-origin text pairs.

Real-world multi-modal datasets include multiple video-text pairs $(V, Q)$, where $V$ denotes the video and $Q$ denotes one of corresponding texts. These texts differ in two levels: word level and sentence structure level. Therefore, we have two types of text rewriting by a two-step rewriting process: word-level rewriting and structure-level rewriting. For convenience, we take the positive text augmentation as an example.

**Word-level augmentation.** In the first step, we directly rewrite the original text by changing some words. With pre-trained LLMs, we can rewrite all the texts by the following prompt: $P'_{i,m} \leftarrow$ LLM($Q_m$, "Rewrite the text '**text**' concisely by changing the $i$-th word while keeping the meaning"), where **text** is substituted with the given text, and we utilize the underlined text to prompt our model for producing morphologically diverse text expressions.

To evaluate the significance of $q_i$ on the semantics of $Q_m$, we can evaluate the semantic change before and after removing this word:

$$S_1(q_i, Q_m, c) = 1 - \cos(c \cup Q_m, c \cup Q_m \setminus \{q_i\}), \tag{1}$$

where $c$ denotes the prompt and $\cos(\cdot, \cdot)$ is the cosine similarity function. In real-world applications, larger $S_1(q_i, Q_m, c)$ denotes that removing $q_i$ will lead to significant semantic changing, indicating that $q_i$ is more relevant.

**Structure-level augmentation.** Since different users tend to utilize various text structures for video grounding, we need to augment the text structure for more diverse texts. Similarly, we utilize the following prompt for structure-level rewriting: $P_{i,m} \leftarrow$ LLM($P'_{i,m}$, "Rewrite the text '**text**' concisely by changing the text structure while keeping the meaning").

Given a sentence $Q_m$, we define the sentence-level relevance of $Q_m^i$ as the probability-weighted semantic similarity with other sentences:

$$S_2(Q_m^i, Q_m^j, c) = \sum_{j=1, j \neq i} \cos(Q_m^i, Q_m^j) p(Q_m^j | c), \tag{2}$$

where $p(Q_m^j, c)$ denotes the generative probability that provides more confidence to $Q_m^j$, and higher $p(Q_m^j, c)$ makes $Q_m^j$ more acceptable. An intuitive observation is that if a sentence is semantically consistent with other sentences, the sentence is more convincing and more representative.

Similar to positive text augmentation in equation 1 and equation 2, we generate negative texts by changing their words and sentence structure. Thus, based on the multi-level language rewriting, we can conduct coarse-grained language alignment.

### 3.2 ATTRIBUTE-BASED TEXT REASONING FOR FINE-GRAINED LANGUAGE ALIGNMENT

In fact, Section 3.1 only considers the semantics of the sentence itself, ignoring the latent information of the sentence. For example, "a person is driving a car" contains two significant objects: "person" and "car". "person" corresponds to the following attributes: a head, two eyes, two arms, etc, while the attributes of "car" include: four wheels, a steering wheel, etc. There attributes will assist VLMs to understand videos and texts for bridging the visual and textual gap.

**Attribute generation.** For some semantically similar sentences, they always have similar attributes. Therefore, we generate the attributes for all positive and negative texts. Although embedding attributes can help us to understand the sentence, current VLM models cannot fully understand the latent semantics. For example, "a person is driving a car" and "a car is running on the road" have similar semantics. Therefore, rather than directly using the original sentence, we design a model with high confidence in visual attributes. Two intuitions are considered in this model: 1) different from the original sentence, aligning explicitly with visual attributes can push the deigned model to mine the inherent semantics in the given sentence. 2) visual attributes contain more fine-grained features, which can provide more details for cross-modal reasoning.

Firstly, we utilize video and text encoders to extract the video and text features. Since our framework is plug-and-play, it does not depend on specific feature encoders. For the fair comparison, we adopt the same video and text encoders with compared methods. For the text $Q$ with $J$ words, we denote word-level text feature as $f^W = \{f_j^w\}_{j=1}^J \in \mathbb{R}^{J \times d}$ and the sentence-level text feature as $f^q \in \mathbb{R}^d$, where $d$ is feature dimension. Similarly, we denote the extracted video features as $f^V = \{f_i^v\}_{i=1}^{N_v} \in \mathbb{R}^{N_v \times d}$, where $N_v$ is the frame number.

**Attribute sampling.** We find that some generated attributes have a stronger semantic correlation with visual features than others, and some attributes have less significance, which will lead to high computational cost. Therefore, removing some

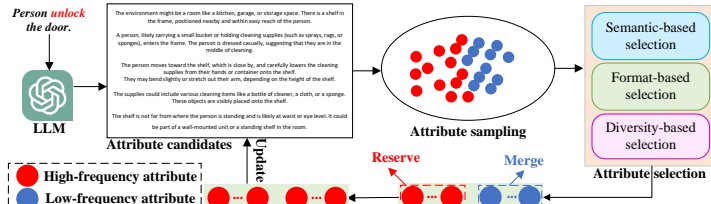

Figure 3: Our attribute selection module.

low significance can not only decrease the computational cost but also improve the model generalization. As shown in Figure 3, we address the problem by selecting effective attributes from an attribute pool. Two main criteria are utilized during the attribute selection: Firstly, we prioritize attributes that are both representative and non-redundant. Secondly, we seek attributes with the highest semantic relevance to the images when compared to other attributes. Finally, we use the following steps for attributes: 1) For the attributes $a_m$ associated with sentence $Q$, we partition them into $N_c$ clusters based on their feature similarity. This clustering strategy aims to ensure that each cluster represents a distinct aspect, e.g., color or shape, in the descriptions. 2) In each cluster, we rank the attributes by assessing their similarity to visual features, and select the one with the highest relevance. By the above strategy, the following attributes will be filtered out: non-visual attributes and incorrect visual attributes that are semantics-unrelated to the videos. To obtain the optimal attributes, we introduce the following attribute selection strategies:

**Semantic-based selection.** Firstly, we want to make the sentence text has the similar semantics with its generated attributes. Since the NLI model can mine the relationship between texts and the attributes by inferring the logical entailment, we introduce an NLI-based binary filter ($f_{nli}$) as a critic, and discard the pairs which do not achieve the entailment score over the threshold $\gamma_1$:

$$O_1(x, y) = \mathbb{1}\{f_{nli}(x \Rightarrow y) \geq \gamma_1\},$$

where $x$ denotes the input, and $y$ means the output.

**Format-based selection.** When we rewrite the given sentence, we need to make the format of the given sentence various, and preserve its original meaning. Thus, we want to filter the origin-variance pair to learn the format-free dissimilarity. Especially, two metrics are used to evaluate the dissimilarity: 1) the token overlap between different sentences and 2) their syntactic difference. For the first, we filter the pairs with a higher Rouge-L (Lin, 2004) than a threshold $\gamma_3$. As for the syntactic difference, we first parse the constituency tree of the origin and variety, and then filter the pairs based on their tree edit distance:

$$O_2(x, y) = \mathbb{1}\{D_t(x, y) \geq \gamma_2 \ \wedge f_{rou}(x, y) \leq \gamma_3 \},$$

where $D_t(\cdot, \cdot)$ denotes the tree edit distance. In equation 3.2, the two dimensions of dissimilarity complement each other. On the one hand, $f_{rou}(\cdot, \cdot)$ promotes lexical divergence in each pair. On the other hand, $D_t(\cdot, \cdot)$ can be used to preempt "hacking" the word-overlap metric by simply switching a few words in the source sentence with corresponding synonyms.

**Diversity-based selection.** For sentence rewriting, we need a diverse range of generated sentences since the diversity of attributes can directly affect the robustness of the trained model. Therefore, we introduce a critic $O_3$ for the diversity. We define two pairs $(x_1, y_1)$ and $(x_2, y_2)$ to be duplicates when one pair entails another, either on the input side ($x_1 \Rightarrow x_2$) or on the output side ($y_1 \Rightarrow y_2$). In the diversity filter, we first cluster all entailing pairs, and then discard all but one with the largest entailment score. Thus, we can utilize the graph traversal for the diversity filter.

Based on the above critics, we can filter the attribute candidate pool $\mathcal{A}$ into an updated pool $\mathcal{U}$:

$$\mathcal{U} = \{(x, y) | (x, y) \in \mathcal{A}, O_1 \wedge O_2 \wedge O_3(x, y) = 1\}.$$

### 3.3 Weighted Sentence Incorporation for Cross-modal Fusion

In fact, different words (*e.g.*, noun, verb, and adjective) have distinct significance in text understanding. For instance, some adjectives are more important for video grounding in some cases, while some verbs are more significant for distinguishing different target moments. Previous VLM methods treat all sentence components equally, which might limit these methods to fully understand the entire sentence. For example, if there is no adjective in the anchor text, the negative text with adjectives cannot contribute to our model since the adjective is not discriminative for the text. Thus, we aim to analyze the relative significance of each word to adaptively integrate different words, where we adaptively predict the salience of sentence components for each anchor text. Without any supervision, we can obtain the significance score which means which word is more significant for text understanding. By the module, we can find an optimal integration strategy of sentence components, which makes VLM selectively understand different sentence components for a given text.

**Incorporating generated sentences.** Based on these positive and hard negative samples, we can encourage the designed VLM models to distinguish the difference between different words in each sentence part. For supervising the VLM model to understand the text input, we introduce a contrastive loss based on three types of text input:

$$\mathcal{L}_{cl}^i = -\log \frac{\beta \cdot \exp[1/\tau \cdot \cos(f^V, g_i^n)]}{(1 - \beta) \cdot \exp[1/\tau \cdot \cos(f^V, g_i^{p_i})] + \beta \cdot \exp[1/\tau \cdot \cos(f^V, g_i^n)]}, \quad (3)$$

where $\beta \in (0, 1)$ is a parameter; $g_i$ denotes the $i$-th text; $g_i^{n_{i,j}}$ and $g_i^{p_i}$ denote the the negative text and the positive text, respectively; $\tau$ denotes the temperature parameter. By equation 3, we can enhance the effectiveness of the designed model by these generated auxiliary texts.

**Weighted contrastive loss.** Since the visual features have higher computational complexity, we generate the positive and negative texts only by the original text (*i.e.*, anchor text) without considering the video input. Since different words contribute variously to sentence understanding, we target to find the most discriminative word for better text understanding by the following loss:

$$\mathcal{L}_{CL}^i = \max(\mathcal{L}_{cl}^{i,1}, \mathcal{L}_{cl}^{i,2}, \ldots, \mathcal{L}_{cl}^{i,C}). \quad (4)$$

For $C$ contrastive losses ($\mathcal{L}_{cl}^{i,1}, \ldots, \mathcal{L}_{cl}^{i,C}$), each contrastive loss computed by equation 3 corresponds to a specific negative text, where the corresponding sentence component is changed. In equation 4, the maximum of these decomposed losses corresponds to the sentence component that is most clearly identified. Considering the significance score in equation 1 and equation 2, we can obtain the finally weighted contrastive loss as follows:

$$\mathcal{L}_{weighted} = \sum\nolimits_{i,j,m,c} S_1(q_i, Q_m, c) \cdot S_2(Q_m^i, Q_m^j, c) \cdot \mathcal{L}_{CL}^i. \quad (5)$$

Since our method is plug-and-play, we borrow the cross-modal fusion module from an open-source works into our framework, which is the base version of our method.

## 4 Exeriments

**Datasets.** For a fair comparison, we utilize the following datasets for evaluation. 1) For the task, we utilize three datasets: ActivityNet Captions (Caba Heilbron et al., 2015), and Charades-STA (Sigurdsson et al., 2016) and TACoS (Regneri et al., 2013). 2) For the VTR task, we adopt two datasets: MSRVTT (Xu et al., 2016) and LSMDC. 3) For the VideoQA task, we use two datasets: NExT-QA (Xiao et al., 2021) and STAR (Wu et al., 2021). More details are placed in Section B.1.

Table 1: Text-to-video and video-to-text retrieval comparisons on MSR-VTT (Xu et al., 2016).

| Method | Without text augmentation | | | | | With text augmentation | | | | |
|---|---|---|---|---|---|---|---|---|---|---|
| | R@1↑ | R@5↑ | R@10↑ | MdR↓ | MnR↓ | R@1↑ | R@5↑ | R@10↑ | MdR↓ | MnR↓ |
| Text-to-video retrieval | | | | | | | | | | |
| *CLIP-ViT-B/32* | | | | | | | | | | |
| X-Pool (Gorti et al., 2022) | 46.9 | 72.8 | 82.2 | 2.0 | 14.3 | 40.1 | 68.2 | 76.5 | 4.0 | 18.9 |
| **+Ours** | **47.8** | **74.9** | **83.5** | **1.0** | **12.3** | **45.9** | **72.3** | **81.4** | **2.0** | **13.6** |
| CLIP-ViP (Xue et al., 2023) | 50.1 | 74.8 | 84.6 | 1.0 | – | 42.3 | 69.4 | 77.8 | 3.0 | 16.5 |
| **+Ours** | **51.7** | **75.3** | **85.9** | **1.0** | **11.8** | **50.4** | **73.6** | **84.5** | **1.0** | **12.4** |
| T-MASS (Wang et al., 2024) | 50.2 | 75.3 | 85.1 | 1.0 | 11.9 | 42.1 | 68.9 | 79.2 | 2.0 | 15.8 |
| **+Ours** | **52.3** | **77.9** | **87.6** | **1.0** | **10.9** | **51.4** | **70.2** | **81.3** | **1.0** | **11.7** |
| *CLIP-ViT-B/16* | | | | | | | | | | |
| X-Pool (Gorti et al., 2022) | 48.2 | 73.7 | 82.6 | 2.0 | 12.7 | 39.7 | 68.5 | 78.4 | 4.0 | 16.5 |
| **+Ours** | **50.7** | **76.2** | **85.2** | **1.0** | **12.4** | **48.9** | **75.3** | **84.0** | **1.0** | **13.8** |
| CLIP-ViP (Xue et al., 2023) | 54.2 | 77.2 | 84.8 | 1.0 | – | 51.2 | 73.9 | 80.4 | 2.0 | 14.8 |
| **+Ours** | **56.8** | **79.4** | **85.9** | **1.0** | **11.3** | **53.6** | **77.8** | **84.2** | **1.0** | **12.5** |
| T-MASS (Wang et al., 2024) | 52.7 | 77.1 | 85.6 | 1.0 | 10.5 | 49.2 | 70.5 | 83.9 | 2.0 | 16.7 |
| **+Ours** | **54.9** | **82.6** | **86.8** | **1.0** | **10.2** | **53.4** | **81.0** | **86.2** | **1.0** | **11.5** |
| Video-to-text retrieval | | | | | | | | | | |
| *CLIP-ViT-B/32* | | | | | | | | | | |
| X-Pool (Gorti et al., 2022) | 44.4 | 73.3 | 84.0 | 2.0 | 9.0 | 41.2 | 68.5 | 80.4 | 3.0 | 13.8 |
| **+Ours** | **45.8** | **76.4** | **87.3** | **1.0** | **7.5** | **42.7** | **74.5** | **86.0** | **2.0** | **8.3** |
| UATVR (Fang et al., 2023) | 46.9 | 73.8 | 83.8 | 2.0 | 8.6 | 43.0 | 67.9 | 78.3 | 3.0 | 11.7 |
| **+Ours** | **49.7** | **75.6** | **86.4** | **1.0** | **7.3** | **47.8** | **74.0** | **83.9** | **2.0** | **7.8** |
| T-MASS (Wang et al., 2024) | 47.7 | 78.0 | 86.3 | 2.0 | 8.0 | 42.9 | 73.5 | 82.6 | 3.0 | 13.9 |
| **+Ours** | **51.5** | **79.9** | **89.8** | **1.0** | **6.4** | **49.5** | **78.1** | **87.5** | **1.0** | **8.2** |
| *CLIP-ViT-B/16* | | | | | | | | | | |
| X-Pool (Gorti et al., 2022) | 46.4 | 73.9 | 84.1 | 2.0 | 8.4 | 42.8 | 72.0 | 81.6 | 3.0 | 10.5 |
| **+Ours** | **50.2** | **77.4** | **86.3** | **2.0** | **6.1** | **48.7** | **76.0** | **84.2** | **2.0** | **7.5** |
| UATVR (Fang et al., 2023) | 48.1 | 76.3 | 85.4 | 2.0 | 8.0 | 41.6 | 73.0 | 81.9 | 3.0 | 10.6 |
| **+Ours** | **50.9** | **77.4** | **90.5** | **2.0** | **6.8** | **48.9** | **76.3** | **87.9** | **2.0** | **7.6** |
| T-MASS (Wang et al., 2024) | 50.9 | 80.2 | 88.0 | 1.0 | 7.4 | 48.3 | 75.6 | 84.9 | 2.0 | 8.9 |
| **+Ours** | **53.7** | **84.2** | **91.5** | **1.0** | **3.4** | **50.8** | **82.7** | **90.4** | **1.0** | **4.9** |

Table 2: VideoQA performance comparison on NExT-QA dataset, where the value means the accuracy of providing the right answer.

| Method | # Frames | Without text augmentation | | | With text augmentation | | |
|---|---|---|---|---|---|---|---|
| | | Temporal | Causal | Description | Temporal | Causal | Description |
| All-in-One (Wang et al., 2023) | 32 | 48.6 | 48.0 | 63.2 | 40.2 | 37.9 | 53.8 |
| **+Ours** | **32** | **50.1** | **51.9** | **64.7** | **48.6** | **50.2** | **61.3** |
| Just Ask (Yang et al., 2021a) | 20 | 51.4 | 49.6 | 63.1 | 42.7 | 40.1 | 54.0 |
| **+Ours** | **20** | **54.3** | **52.9** | **67.8** | **50.9** | **49.3** | **62.7** |
| MIST (Gao et al., 2023) | 32 | 56.6 | 54.6 | 66.9 | 51.9 | 48.2 | 55.3 |
| **+Ours** | **32** | **60.3** | **56.9** | **69.8** | **57.2** | **55.4** | **67.9** |
| HiTeA (Ye et al., 2022) | 16 | 58.3 | 62.4 | 75.6 | 52.2 | 57.6 | 59.3 |
| **+Ours** | **16** | **62.8** | **65.7** | **77.3** | **60.4** | **63.9** | **74.9** |
| InternVideo (Wang et al., 2022a) | 8 | 58.5 | 62.5 | 75.8 | 52.9 | 57.4 | 70.3 |
| **+Ours** | **8** | **62.5** | **66.3** | **76.4** | **61.8** | **59.7** | **74.5** |
| BLIP-2 (Li et al., 2023b) | 4 | 67.2 | 70.3 | 79.8 | 64.0 | 61.9 | 72.3 |
| **+Ours** | **4** | **70.1** | **72.9** | **80.4** | **69.2** | **70.1** | **78.4** |
| SeViLA (Yu et al., 2023) | 4 | 67.7 | 72.1 | 82.2 | 64.0 | 66.8 | 76.9 |
| **+Ours** | **4** | **72.4** | **74.9** | **85.3** | **70.5** | **72.7** | **83.9** |

Table 3: Comparison Results on STAR VideoQA benchmark.

| Method (Frames Number) | Without text augmentation | | | | With text augmentation | | | |
|---|---|---|---|---|---|---|---|---|
| | Interaction | Sequence | Prediction | Feasibility | Interaction | Sequence | Prediction | Feasibility |
| All-in-One (Wang et al., 2023) (32) | 47.5 | 50.8 | 47.7 | 44.0 | 42.9 | 48.5 | 44.0 | 40.2 |
| **+Ours (32)** | **48.3** | **51.9** | **49.6** | **45.7** | **47.9** | **51.3** | **48.7** | **44.3** |
| MIST (Gao et al., 2023) (32) | 55.5 | 54.2 | 54.2 | 44.4 | 50.7 | 51.4 | 50.2 | 38.4 |
| **+Ours (32)** | **58.6** | **59.5** | **58.4** | **47.0** | **57.0** | **56.3** | **57.2** | **45.8** |
| InternVideo (Wang et al., 2022a) (8) | 62.7 | 65.6 | 54.9 | 51.9 | 55.6 | 61.0 | 50.3 | 47.2 |
| **+Ours (8)** | **63.8** | **67.7** | **58.9** | **55.2** | **61.8** | **64.9** | **57.4** | **54.3** |
| SeViLA (Yu et al., 2023) (4) | 63.7 | 70.4 | 63.1 | 62.4 | 58.7 | 62.2 | 57.9 | 57.8 |
| **+Ours (4)** | **66.7** | **72.9** | **66.4** | **65.3** | **65.3** | **68.9** | **64.2** | **63.4** |
| BLIP-2 (Li et al., 2023b) (4) | 65.4 | 69.0 | 59.7 | 54.2 | 60.9 | 66.3 | 54.3 | 50.1 |
| **+Ours (4)** | **67.8** | **72.5** | **61.4** | **56.8** | **66.2** | **71.6** | **58.7** | **55.3** |

**Evaluation metrics.** For the VTR task, we utilize Recall at rank $\{1, 5, 10\}$ (R@1, R@5, and R@10), Median Rank (MdR), and Mean Rank (MnR) for evaluating the retrieval performance. For the VSG task, we evaluate the grounding performance by "R@n, IoU=m", which means the percentage of queries having at least one result whose Intersection over Union (IoU) with ground truth is larger than m. In our experiments, we use $n \in \{1, 5\}$ for all datasets, $m \in \{0.5, 0.7\}$ for ActivityNet Captions and Charades-STA, $m \in \{0.3, 0.5\}$ for TACoS. As for the VideoQA task, we introduce the following metrics: temporal, causal, description, interaction, sequence, prediction

Table 4: VSG performance comparison under official train/test splits, where "FS" denotes "fully-supervised" and "WS" means "weakly-supervised".

| Method | Type | Without text augmentation | | | | With text augmentation | | | |
|---|---|---|---|---|---|---|---|---|---|
| | | R@1, IoU=0.3 | R@1, IoU=0.5 | R@5, IoU=0.3 | R@5, IoU=0.5 | R@1, IoU=0.3 | R@1, IoU=0.5 | R@5, IoU=0.3 | R@5, IoU=0.5 |
| ActivityNet Captions | | | | | | | | | |
| 2D-TAN | FS | 59.45 | 44.51 | 85.53 | 77.13 | 48.32 | 29.38 | 71.36 | 62.30 |
| +Ours | **FS** | **60.46** | **45.29** | **87.94** | **77.43** | **51.86** | **32.64** | **72.98** | **63.75** |
| MMN | FS | 65.05 | 48.59 | 87.25 | 79.50 | 55.30 | 31.76 | 74.88 | 71.52 |
| +Ours | **FS** | **66.05** | **49.31** | **89.75** | **81.27** | **58.76** | **33.08** | **75.33** | **73.59** |
| G2L | FS | - | 51.68 | - | 81.32 | 55.75 | 33.01 | 75.25 | 70.89 |
| +Ours | **FS** | **66.34** | **54.26** | **91.77** | **84.29** | **60.90** | **46.86** | **84.39** | **80.62** |
| VCA | WS | 50.45 | 31.00 | 71.79 | 53.83 | 31.74 | 25.37 | 46.98 | 42.76 |
| +Ours | **WS** | **51.72** | **33.19** | **72.85** | **55.11** | **32.99** | **28.56** | **48.31** | **44.07** |
| WSTAN | WS | 52.45 | 30.01 | 79.38 | 63.42 | 33.72 | 25.74 | 49.30 | 45.88 |
| +Ours | **WS** | **53.10** | **31.56** | **80.24** | **65.77** | **35.20** | **27.99** | **51.84** | **48.69** |
| CNM | WS | 55.68 | 33.33 | - | - | 35.72 | 28.95 | 50.06 | 48.72 |
| +Ours | **WS** | **56.11** | **34.08** | **81.09** | **67.34** | **39.56** | **31.77** | **52.88** | **51.99** |
| Charades-STA | | | | | | | | | |
| 2D-TAN | FS | 39.81 | 23.25 | 79.33 | 52.15 | 20.18 | 11.35 | 47.05 | 33.82 |
| +Ours | **FS** | **40.27** | **24.95** | **82.96** | **53.28** | **23.99** | **14.75** | **49.22** | **34.18** |
| MMN | FS | 47.31 | 27.28 | 83.74 | 58.41 | 25.33 | 18.80 | 45.97 | 35.08 |
| +Ours | **FS** | **49.07** | **29.32** | **85.06** | **60.13** | **26.87** | **22.48** | **46.03** | **37.85** |
| G2L | FS | 47.91 | 28.42 | 84.80 | 59.33 | 26.54 | 19.85 | 48.06 | 36.70 |
| +Ours | **FS** | **55.77** | **32.97** | **91.38** | **60.39** | **34.85** | **27.96** | **74.28** | **46.70** |
| VCA | WS | 38.13 | 19.57 | 78.75 | 37.75 | 17.87 | 12.39 | 45.70 | 22.13 |
| +Ours | **WS** | **40.95** | **20.31** | **80.42** | **39.26** | **18.63** | **15.72** | **46.17** | **23.88** |
| WSTAN | WS | 29.35 | 12.28 | 76.13 | 41.53 | 8.15 | 5.43 | 35.27 | 11.86 |
| +Ours | **WS** | **30.24** | **14.06** | **77.35** | **42.99** | **10.77** | **6.92** | **37.40** | **13.88** |
| CNM | WS | 35.15 | 14.95 | - | - | 14.34 | 9.65 | 43.88 | 18.79 |
| +Ours | **WS** | **35.72** | **16.33** | **76.52** | **43.18** | **16.83** | **12.05** | **45.60** | **21.64** |

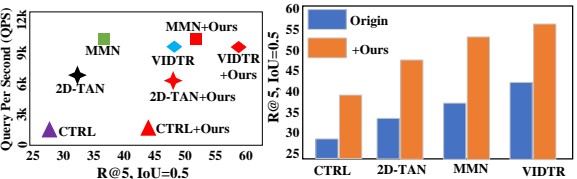

| Method | Run-time | Model Size | R@1, IoU=0.5 | Aug time |
|---|---|---|---|---|
| ACRN | 6.88s | 128M | 14.62 | - |
| CTRL | 1.24s | **22M** | 13.30 | - |
| TGN | 1.95s | 166M | 18.90 | - |
| 2DTAN | 1.72s | 232M | 25.32 | - |
| DRN | 1.31s | 214M | 23.17 | - |
| MomentDiff | 2.40s | 248M | 33.68 | - |
| **G2L+Ours** | **0.95s** | 113M | **40.31** | 0.20s |

Table 5: Figure: Performance comparison with state-of-the-art methods on the TACoS for the VSG task, where left figure compares the effectiveness (R@5, IoU=0.5) and the efficiency (QPS), right figure shows that our method can serve as a plug-and-play module to enhance their efficiency. Table: Efficiency comparison for VSG on TACoS without text augmentation. "Aug time" denotes the time of generating multi-level texts.

Table 6: Main ablation study on the VSG task with G2L as the base model, where we remove each key individual component to investigate its effectiveness.

| Model | ActivityNet Captions | | | | Charades-STA | | | |
|---|---|---|---|---|---|---|---|---|
| | R@1 IoU=0.3 | R@1 IoU=0.5 | R@5 IoU=0.3 | R@5 IoU=0.5 | R@1 IoU=0.5 | R@1 IoU=0.7 | R@5 IoU=0.5 | R@5 IoU=0.7 |
| Ours(a) | 53.77 | 40.28 | 76.94 | 72.25 | 28.51 | 20.34 | 67.85 | 38.71 |
| Ours(b) | 55.35 | 42.03 | 79.50 | 74.91 | 30.88 | 23.92 | 70.66 | 41.58 |
| Ours(c) | 57.63 | 43.86 | 81.34 | 77.99 | 32.50 | 24.03 | 71.76 | 42.92 |
| **Ours(full)** | **60.90** | **46.86** | **84.39** | **80.62** | **34.85** | **27.96** | **74.28** | **46.70** |

and feasibility. In these metrics, lower MdR and MnR denotes better performance. For the metrics, higher value means better performance. Bold denotes the best performance.

**Implementation details.** For video encoding, we utilize the $112 \times 112$ pixels shape of every frame of videos. As for the text encoder, we feed the texts to a pre-trained txt encoder to embed word-level features. The dimensions $d$ of video and text tokens are 512. We set $\gamma_1 = 0.8, \gamma_2 = 0.6, \gamma_3 = 0.7$ and $\mu = 0.6$ in our experiments to achieve the best performance. We train our model for 100 epochs with an Adam optimizer with the learning rate $3 \times 10^{-4}$.

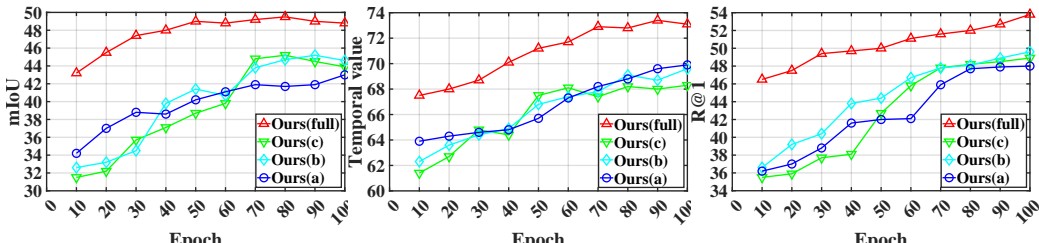

Figure 4: Training performance of each ablation module with text augmentation on the ActivityNet Captions dataset (left, VSG), the NexT-QA dataset (middle, VideoQA) and the MSR-VTT dataset (right, VTR).

## 4.1 PERFORMANCE COMPARISON

Following previous open-source methods, we directly cite the corresponding results from compared methods. In this paper, we treat our as the plug-and-play module for state-of-the-art VLM models to improve their performance.

**Performance comparison on the VTR task.** VTR is a challenging multi-modal task, which requires the designed model can effectively bridge the gap between videos and texts. In this paper, we consider two subtask: text-to-video retrieval and video-to-text retrieval. Table 1 illustrates the effectiveness of our model as the plug-and-play module for previous VTR methods. We can find that when using augmented text, all the compared methods suffer performance degradation. The core reason is that previous VTR methods pay less attention to the language input, and ignore much language information in the sentence query. By using our model as the plug-and-play module, previous method can obtain significant performance improvement since our proposed model can fully mine latent language semantics.

**Performance comparison on the VideoQA task.** Similar to the VTR task, we conduct performance comparison VideoQA performance comparison. The experimental results are summarized in Table 2 and Table 3, where the performance of previous methods was unsatisfactory. The key reason is that previous methods have difficulty in understanding the rewritten question. Different from them, we can explore more deep and fine-grained language information by attribute-based text reasoning.

**Performance comparison on the VSG task.** We conduct VSG performance comparison on all three datasets with official train/test splits under both fully-supervised (Gao et al., 2017; Li et al., 2023a; Liu et al., 2018; Li et al., 2023c; Yuan et al., 2019a; Zhang et al., 2019b; 2020b; Zeng et al., 2020; Gao & Xu, 2021; Zhang et al., 2021; Gao et al., 2021; Wang et al., 2022b) and weakly-supervised setting (Chen et al., 2022; Yang et al., 2021b; Zhang et al., 2020c; Wang et al., 2021b;a; Zheng et al., 2022). Table 4 and 5 reports the quantitative comparison results. Obviously, our proposed model can help state-of-the-art VSG methods for performance improvement over all metrics on three datasets, which demonstrates the superiority of our proposed model. It is mainly because our model can fully understand the query knowledge by the text augmentation process.

**Efficiency comparison.** We evaluate the efficiency of our proposed model, by fairly comparing its running time and model size in the inference phase with existing open-source methods for the VSG task on TACoS. As shown in Table 5, it can be observed that we achieve much faster processing speeds with relatively fewer learnable parameters.

## 4.2 ABLATION STUDY AND ANALYSIS

**Main ablation studies.** To demonstrate the effectiveness of each component in our model, we conduct ablation studies regarding the components (*i.e.*, Augmenting texts by Pre-trained LLMs, Generating Positive and Negative texts, Significance Estimation for Sentence Component Integration and Cross-modal Fusion) in Table 6. In particular, we remove each key individual module to investigate its contribution. For convenience, we design four ablation models: 1) Ours(a). We remove the "Augmenting texts by Pre-trained LLMs" module while keeping the other three modules. 2) Ours(b). We remove the "Generating Positive and Negative texts" module while keeping the other three modules. 3) Ours(c). We remove the "Significance Estimation for Sentence Component Integration" module while keeping the other three modules. Besides, we use our full model as the

Table 7: Ablation study on different word types for the text-to-video task on DiDeMo (Anne Hendricks et al., 2017) and VATEX (Wang et al., 2019), where T-MASS (Wang et al., 2024) is the base model with CLIP-ViT-B/32 as backbone.

| Method | DiDeMo | | | | | VATEX | | | | |
|---|---|---|---|---|---|---|---|---|---|---|
| | R@1↑ | R@5↑ | R@10↑ | MdR↓ | MnR↓ | R@1↑ | R@5↑ | R@10↑ | MdR↓ | MnR↓ |
| w/o Verb | 46.0 | 74.1 | 82.7 | 2.0 | 14.3 | 61.2 | 93.2 | 94.0 | 2.0 | 2.7 |
| w/o Noun | 43.1 | 71.8 | 82.3 | 2.0 | 15.1 | 61.3 | 91.0 | 95.6 | 2.0 | 3.3 |
| w/o Subject | 48.6 | 77.1 | 84.4 | 2.0 | 13.4 | 65.2 | 92.7 | 95.9 | 1.0 | 3.0 |
| **Full model** | **52.7** | **79.3** | **88.6** | **1.0** | **10.4** | **64.9** | **93.7** | **98.2** | **1.0** | **1.4** |

baseline: Ours(full). As shown in Table 6, all four modules contribute a lot to the final performances on all three datasets, demonstrating their effectiveness under the VSG task.

**Training process of different ablation models.** Following (Lin et al., 2020b), we analyze the training process and retrieval performance of different ablation models in Figure 4. We can obtain the following representative observations: (i) During training, Our(full) outperforms other ablation models, which further demonstrates the effectiveness of each module. (ii) Our(full) converges faster than ablation models, showing that our full model is more efficient. For instance, Our(full) converges within 70 epochs, while Our(c) converges after 80 epochs. Thus, our full model can process these challenging datasets more efficiently.

**Analysis on different word types.** During generating negative texts, we change a part of the sentence to help us understand the whole sentence. As shown in Table 4.2, we analyze the effect of different word types. Among three word types (noun, verb, and subject), the noun is the most significant. It is because the noun can help our model localize the object in the given video for text-to-video task. Besides, the noun can be used to generate more semantic-rich attributes for fine-grained language alignment. On the contrary, the subject brings minimal performance improvement.

**Visualization** Figure 5 depicts the grounding visualizations. Our model can significantly improve the state-of-the-art

**Q1:** *Person pours water into a glass.*
**Q2:** *Water is poured into a glass.*

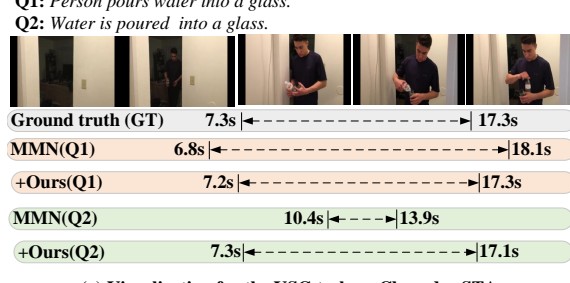

(a) Visualization for the VSG task on Charades-STA.

**Q1:** *Kids are in a classroom finger painting.*
**Q2:** *Children are painting in room.*

(b) Visualization for the VTR task on MSR-VTT.

**Q1:** *How did the woman in yellow support the boy in blue at the start?*
**Q2:** *How did the woman in yellow assist the boy in blue at the beginning?*

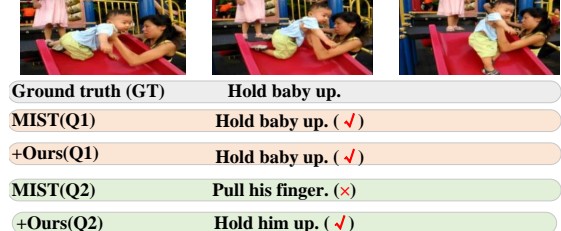

(c) Visualization for the VideoQA task on NexT-QA.

Figure 5: Visualization results.

VLM methods for different tasks. This is because our model can fully understand the textual input by attribute-based text reasoning.

## 5 CONCLUSION

In this paper, we rethink the LLM task from the user-friendly language input. We observe that many VLMs cannot fully understand the language texts. Given some texts with similar semantics and a video, these VLMs output various results. Thus, we design a plug-and-play framework to improve the generation ability of previous methods on various text templates. Extensive experiments on many challenging datasets show that our framework can serve as the plug-and-play module for state-of-the-art VLM works to improve their performance on various video-language tasks. In our future work, we will extend our model into more multi-modal tasks.

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

## A APPENDIX

### A.1 AUGMENTING TEXTS BY PRE-TRAINED LLMS

Considering the strong natural language processing of pre-trained LLMs, we will generate various text augmentations by pre-trained LLMs. Inspired by the effectiveness of LLMs, we rewrite the texts in VLM datasets to generate variation-origin pairs based on the following approaches.

#### A.1.1 AUGMENTATION BY HUMAN.

To make our model more user-friendly, we randomly invite ten persons from different countries to rewrite some texts. For convenience, we randomly choose some video-text pairs from these VLM datasets. For obtaining diverse text variations, we encourage rewriters to rewrite these texts based on the target video segments. The human-based rewriting approach will enhance the creation of text augmentation. Finally, we can obtain the variation-origin text pairs that include the original texts and the corresponding human-rewritten version.

**Augmentation by chat robots.** Recently, LLMs-based chat robots (*e.g.*, ChatGPT or Bard) have achieved impressive performance in natural language processing. Thus, we target to rewrite the texts by chat robots. Firstly, we randomly choose some texts from the VSG datasets. Then, we utilize the web portals of chat robots to generate target texts by providing prompts. Some examples of the chat robot rewriting is shown in our supplementary material. With the powerful language processing ability of these chat robots, we can rewrite texts by utilizing different templates and vocabularies. The rewriting approach can preserve most textual semantics corresponding to the target segment.

**Augmentation by open-source LLMs.** Since it will lead to significant financial and time costs if we generate all the augmented texts by these closed-source chat robots (*e.g.*, ChatGPT and Bard), we utilize an open-source LLMs, LLaMA, to generate the positive texts. Due to the strong generalization ability, LLaMA can be directly used to rewrite all the texts in the video-text datasets. For better generation ability, we leverage the LLaMA-7B model for text generation to guarantee that the generated texts are diverse and semantically relevant to the original texts.

Based on the above approaches, we can obtain three types of variation-origin text pairs: Bard-based, ChatGPT-based, and LLaMA-based. Then, we treat them as inputs for the In-Context Learning strategy (Zhang et al., 2024). In each approach, we randomly choose some texts from the video-text datasets for target text generation, which will generate some variation-origin text pairs. These pairs contain comprehensive and diverse training samples as the input of our framework.

### A.2 MORE DETAILS FOR RELATED WORKS

**Fully-supervised VSG.** VSG is a new task introduced recently (Gao et al., 2017; Anne Hendricks et al., 2017). Most previous algorithms (Anne Hendricks et al., 2017; Gao et al., 2017; Chen et al., 2018; Zhang et al., 2019b; Yuan et al., 2019a; Zhang et al., 2020b; Liu et al., 2021) have been proposed within the propose-and-rank framework, which first generates moment candidates and then utilizes multimodal matching to retrieve the most relevant candidate for a text. Some of them (Anne Hendricks et al., 2017; Gao et al., 2017) take multiple sliding windows as candidates. To improve the quality of the candidates, (Zhang et al., 2019b; Yuan et al., 2019a) pre-cut the video on each frame by multiple pre-defined temporal scales, and directly integrate sentence information with fine-grained video clips for scoring. For instance, Xu *et al.* (Xu et al., 2019) introduce a multi-level model to integrate visual and textual features earlier and further re-generate texts as an auxiliary task. Zhang *et al.* (Zhang et al., 2019a) model relations among candidate moments produced from a convolutional neural network with the guidance of the text information. Although these methods achieve great performance, they are severely limited by the heavy computation on proposal matching/ranking, and sensitive to the quality of pre-defined proposals. Recently, many methods (Chen et al., 2020; Yuan et al., 2019b; Zeng et al., 2020; Zhang et al., 2020a; Nan et al., 2021) propose to utilize the boundary-regression framework. Specifically, they directly predict two probabilities at each frame by leveraging cross-modal interactions between video and text, which indicate whether this frame is a start/end frame of the ground truth video moment.

**Weakly-supervised VSG.** Despite the decent progress on the grounding performance, fully-supervised methods severely rely on the numerous annotations, which are significantly labor-intensive and time-consuming to obtain. To alleviate this dense reliance to a certain extent, some weakly-supervised VSG methods are proposed (Mithun et al., 2019; Chen et al., 2019; Lin et al., 2020a). For a weakly supervised VSG task, (Duan et al., 2018) decomposes it into two sub-tasks: event captioning and text localization. (Duan et al., 2018) first assumes that each caption describes only one temporal moment, and then designs a cycle network to train the model. As the pioneering work for weakly-supervised VSG, (Mithun et al., 2019) learns a joint representation between the video and the text by proposing Text-Guided-Attention network and utilizing an attention weight. To improve the exploration and exploitation, (Lin et al., 2020a) chooses the top-K proposals and measures the semantic similarity between the video and the text for localization. By proposing a Semantic Completion Network, (Lin et al., 2020a) treats the masked text as input and predicts the masked words from the video features.

## B MORE EXPERIMENTS

### B.1 MORE DATASET DETAILS

For convenience, we only utilize three datasets on the VSG task as example. We can utilize similar process for the other tasks.

1) **ActivityNet Captions** contains 19,209 videos from ActivityNet (Heilbron et al., 2015) with 71,953 textual descriptions. The videos are of diverse and open contents with an average length of 2 minutes, and the annotated segments are significantly various in length, ranging from several seconds to over 3 minutes. Following the public experimental setting (Zhang et al., 2019b), we use 37,417, 17,505, and 17,031 segment-text pairs for training, validation, and testing. For each dataset split (training, validation, and testing), we generate 10,000 positive texts by each rewriting approach (ChatGPT, Bard, and LLaMA) for each split.

2) **Charades-STA** (Gao et al., 2017) is an extension of the Charades dataset (Sigurdsson et al., 2016) with temporal annotations. It contains 9,848 videos with an average length of 30 seconds and mainly focuses on daily indoor activities. There are 12,408 and 3,720 segment-text pairs in the training and

Table 8: Performance comparison on TACoS dataset (with text augmentation) under official train/test splits.

| Method | Type | R@1, IoU=0.3 | R@1, IoU=0.5 | R@5, IoU=0.3 | R@5, IoU=0.5 |
|--------|------|------|------|------|------|
| CTRL | FS | 10.25 | 8.92 | 27.85 | 16.96 |
| ACRN | FS | 11.70 | 10.38 | 28.04 | 18.30 |
| CMIN | FS | 12.87 | 9.96 | 27.99 | 17.43 |
| SCDM | FS | 14.35 | 10.96 | 28.33 | 18.82 |
| DRN | FS | 16.39 | 11.77 | 30.85 | 20.99 |
| 2D-TAN | FS | 16.98 | 13.79 | 33.02 | 21.13 |
| MMN | FS | 19.90 | 15.32 | 34.77 | 23.85 |
| FVSG | FS | 22.30 | 17.85 | 35.23 | 23.07 |
| RaNet | FS | 22.95 | 18.86 | 37.44 | 22.91 |
| G2L | FS | 23.48 | 19.87 | 37.52 | 24.88 |
| MomentDiff | FS | 26.76 | 18.73 | 38.49 | 25.84 |
| **G2L+Ours** | **-** | **37.92** | **27.48** | **46.73** | **35.95** |

Table 9: Effectiveness comparison on ActivityNet Captions dataset (without text augmentation) under official train/test splits.

| Method | Type | R@1, IoU=0.3 | R@1, IoU=0.5 | R@5, IoU=0.3 | R@5, IoU=0.5 |
|--------|------|------|------|------|------|
| CTRL | FS | - | 29.01 | - | 59.17 |
| 2D-TAN | FS | 59.45 | 44.51 | 85.53 | 77.13 |
| DRN | FS | - | 45.45 | - | 77.97 |
| RaNet | FS | - | 45.59 | - | 75.93 |
| MIGCN | FS | - | 48.02 | - | 78.02 |
| MMN | FS | 65.05 | 48.59 | 87.25 | 79.50 |
| G2L | FS | - | 51.68 | - | 81.32 |
| ICVC | WS | 46.62 | 29.52 | 80.92 | 66.61 |
| LCNet | WS | 48.49 | 26.33 | 82.51 | 62.66 |
| VCA | WS | 50.45 | 31.00 | 71.79 | 53.83 |
| WSTAN | WS | 52.45 | 30.01 | 79.38 | 63.42 |
| CNM | WS | 55.68 | 33.33 | - | - |
| **G2L+Ours** | **-** | **66.34** | **54.26** | **91.77** | **84.29** |

testing sets, respectively. Similar to the ActivityNet Captions dataset, we generate 2,000 positive texts by each rewriting approach (ChatGPT, Bard, and LLaMA) for each dataset split.

3) **TACoS** (Regneri et al., 2013) consists of 127 videos with an average length of 7 minutes. These videos are selected from the MPII Cooking Composite Activities video corpus, which contains activities of cooking scenarios, thus lacking diversity. Following the standard split of (Gao et al., 2017), we use 10,146, 4,589, and 4,083 segment-text pairs for training, validation, and testing, respectively. Similar to the ActivityNet Captions dataset, we generate 4,000 positive texts by each rewriting approach (ChatGPT, Bard, and LLaMA) for each dataset split.

Similarly, we conduct the similar augmentation operation in other datasets.

**Effect of negative samples.** To evaluate the effect of the generated negative samples, we conduct the ablation study for the VideoQA task on the NExT-QA dataset. Table 19 shows the results. Obviously, we can utilize the generated negative samples for performance improvement in terms of all the metrics.

**Effect of attribute.** Similarly, we analyze the significance of the attributes in our proposed framework by performing the ablation study in Table 20. In this table, with the attributes, our model achieves the significant performance improvement. It is because the attributes can help our model fully understand the language input by mining latent fine-grained language semantics.

Table 10: Text-to-video comparisons on DiDeMo (Anne Hendricks et al., 2017) and VATEX (Wang et al., 2019). Bold denotes the best performance. "–": result is unavailable.

| Method | DiDeMo Retrieval | | | | | VATEX Retrieval | | | |
|---|---|---|---|---|---|---|---|---|---|
| | R@1 ↑ | R@5 ↑ | R@10 ↑ | MdR ↓ | MnR ↓ | R@1 ↑ | R@5 ↑ | R@10 ↑ | MdR ↓ |
| *CLIP-ViT-B/32* | | | | | | | | | |
| X-Pool (Gorti et al., 2022) | 44.6 | 73.2 | 82.0 | 2.0 | 15.4 | 60.0 | 90.0 | 95.0 | 1.0 |
| **+Ours** | **47.8** | **75.5** | **84.6** | **1.0** | **14.1** | **61.2** | **93.4** | **98.9** | **1.0** |
| UATVR (Fang et al., 2023) | 43.1 | 71.8 | 82.3 | 2.0 | 15.1 | 61.3 | 91.0 | 95.6 | 1.0 |
| **+Ours** | **45.5** | **72.9** | **84.8** | **1.0** | **13.4** | **65.2** | **92.7** | **98.9** | **1.0** |
| T-MASS (Wang et al., 2024) | 50.9 | 77.2 | 85.3 | 1.0 | 12.1 | 63.0 | 92.3 | 96.4 | 1.0 |
| **+Ours** | **52.7** | **79.3** | **88.6** | **1.0** | **10.4** | **64.9** | **93.7** | **98.2** | **1.0** |

Table 11: Performance comparison on TACoS dataset (without text augmentation) under official train/test splits.

| Method | Type | R@1, IoU=0.3 | R@1, IoU=0.5 | R@5, IoU=0.3 | R@5, IoU=0.5 |
|---|---|---|---|---|---|
| CTRL | FS | 18.32 | 13.30 | 36.69 | 25.42 |
| ACRN | FS | 19.52 | 14.62 | 34.97 | 24.88 |
| CMIN | FS | 24.64 | 18.05 | 38.46 | 27.02 |
| SCDM | FS | 26.11 | 21.17 | 40.16 | 32.18 |
| DRN | FS | - | 23.17 | - | 33.36 |
| 2D-TAN | FS | 37.29 | 25.32 | 57.81 | 45.04 |
| MMN | FS | 39.24 | 26.17 | 62.03 | 47.39 |
| FVSG | FS | 41.48 | 29.12 | 64.53 | 50.00 |
| G2L | FS | 42.74 | 30.95 | 65.83 | 49.86 |
| RaNet | FS | 43.34 | 33.54 | 67.33 | 55.09 |
| MomentDiff | FS | 44.78 | 33.68 | - | - |
| **G2L+Ours** | **-** | **53.56** | **40.31** | **71.62** | **62.06** |

**Influence of corse-grained language alignment.** To show the importance of our corse-grained language alignment strategy, we conduct the ablation study for the VideoQA task on the NExT-QA dataset in Table 21. Table 21 illustrates the effectiveness of the corse-grained language alignment strategy.

**Influence of fine-grained language alignment.** To evaluate the effectiveness of the fine-grained language alignment module, we conduct the ablation study for the VideoQA task on the NExT-QA dataset in Table 22. Obviously, our fine-grained language alignment module can effectively improve the performance over all metrics.

**Analysis on the hyper-parameters.** Moreover, we investigate the robustness of the proposed model to different hyper-parameters in Figure 6. We find that we can obtain the best performance when $\gamma_1 = 0.4, \gamma_2 = 0.85, \gamma_3 = 0.25, \beta = 0.6, \tau = 0.3$.

**Feature visualization.** To investigate the feature distributions of the sentences during language alignment, we randomly choose some origin-variation pairs, and show the t-SNE Van der Maaten & Hinton (2008) visualizations of "before language alignment" and "after language alignment" in Figure 7. We can find that there is a large distribution gap between the origin and the variation of "before language alignment".

Table 12: Text-to-video comparisons on MSRVTT (Xu et al., 2016) and LSMDC (Rohrbach et al., 2015). Bold denotes the best performance.

| Method | MSRVTT | | | | | LSMDC | | | | |
|---|---|---|---|---|---|---|---|---|---|---|
| | R@1↑ | R@5↑ | R@10↑ | MdR↓ | MnR↓ | R@1↑ | R@5↑ | R@10↑ | MdR↓ | MnR↓ |
| *CLIP-ViT-B/32* | | | | | | | | | | |
| X-Pool (Gorti et al., 2022) | 46.9 | 72.8 | 82.2 | 2.0 | 14.3 | 25.2 | 43.7 | 53.5 | 8.0 | 53.2 |
| **+Ours** | **47.8** | **74.9** | **83.5** | **1.0** | **12.3** | **26.7** | **45.8** | **55.5** | **7.0** | **50.9** |
| DiffusionRet (Jin et al., 2023) | 49.0 | 75.2 | 82.7 | 2.0 | 12.1 | 24.4 | 43.1 | 54.3 | 8.0 | 40.7 |
| **+Ours** | **51.7** | **77.9** | **84.5** | **1.0** | **11.8** | **25.7** | **45.2** | **55.8** | **7.0** | **38.5** |
| TEFAL (Ibrahimi et al., 2023) | 49.4 | 75.9 | **83.9** | 2.0 | 12.0 | 26.8 | 46.1 | 56.5 | 7.0 | 44.4 |
| **+Ours** | **51.9** | **77.4** | 83.5 | **1.0** | **11.8** | **28.4** | **46.9** | **58.2** | **6.0** | **42.1** |
| CLIP-ViP (Xue et al., 2023) | 50.1 | 74.8 | 84.6 | 1.0 | – | 25.6 | 45.3 | 54.4 | 8.0 | – |
| **+Ours** | **51.7** | **75.3** | **85.9** | 1.0 | 11.8 | **26.8** | **47.6** | **58.5** | **6.0** | 42.3 |
| T-MASS (Wang et al., 2024) | 50.2 | 75.3 | 85.1 | 1.0 | 11.9 | 28.9 | 48.2 | 57.6 | 6.0 | 43.3 |
| **+Ours** | **52.3** | **77.9** | **87.6** | 1.0 | **10.9** | **30.8** | **50.4** | **59.1** | **5.0** | **40.8** |
| *CLIP-ViT-B/16* | | | | | | | | | | |
| X-Pool (Gorti et al., 2022) | 48.2 | 73.7 | 82.6 | 2.0 | 12.7 | 26.1 | 46.8 | 56.7 | 7.0 | 47.3 |
| **+Ours** | **50.7** | **76.2** | **85.2** | **1.0** | **12.4** | **26.9** | **49.5** | **57.4** | **6.0** | **45.0** |
| CLIP-ViP (Xue et al., 2023) | 54.2 | 77.2 | 84.8 | 1.0 | – | 29.4 | 50.6 | 59.0 | 5.0 | – |
| **+Ours** | **56.8** | **79.4** | **85.9** | 82.8 | **1.0** | **32.6** | **51.8** | **60.9** | **3.0** | 38.7 |
| T-MASS (Wang et al., 2024) | 52.7 | 77.1 | 85.6 | 1.0 | 10.5 | 30.3 | 52.2 | 61.3 | 5.0 | 40.1 |
| **+Ours** | **54.9** | **82.6** | **86.8** | 1.0 | **10.2** | **35.1** | **55.7** | **64.8** | **3.0** | **38.9** |

Table 13: Example of LLM-based text generation, where "Origin" denotes the given original text and "Variation" denotes the generated texts. Although the only difference in the sentence structure between original text and negative text is "door" and "closet", they have different semantics. In contrast, positive text that are more distinct from the original text has similar semantics with original text.

| Origin | Variation | Type | Semantically similar? |
|---|---|---|---|
| Person opens the **door**. | Person opens the closet. | Negative | ✗ |
| | Door is opened by person. | Positive | ✔ |

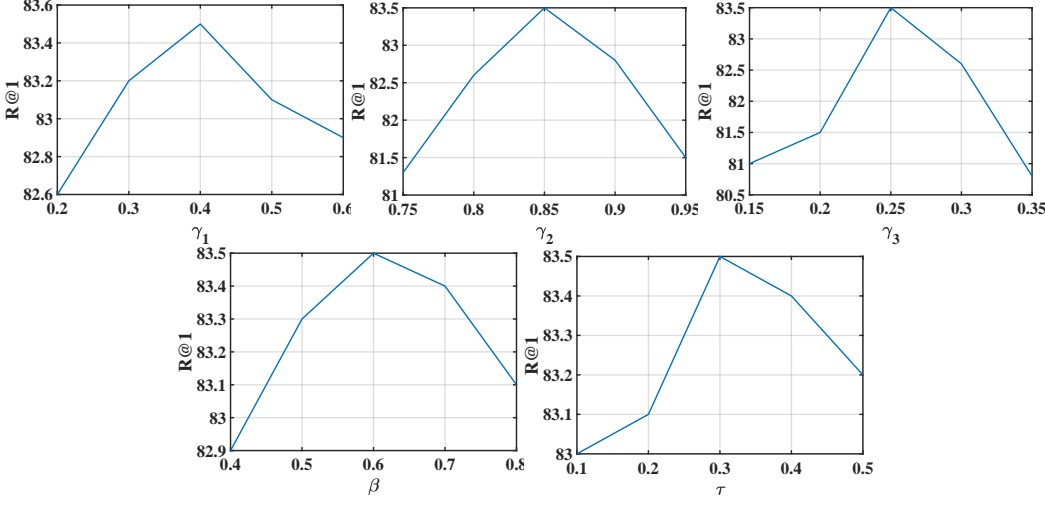

Figure 6: Parameter analysis on the MSR-TT dataset for the text-to-video retrieval task with X-Pool as the base model.

Table 14: Performance comparison on Charades-STA dataset (without text augmentation) under official train/test splits.

| Method | Type | R@1, IoU=0.5 | R@1, IoU=0.7 | R@5, IoU=0.5 | R@5, IoU=0.7 |
|--------|------|--------------|--------------|--------------|--------------|
| VSA-RNN | FS | 10.50 | 4.32 | 48.43 | 20.21 |
| VSA-STV | FS | 16.91 | 5.81 | 53.89 | 23.58 |
| CTRL | FS | 23.62 | 8.89 | 58.92 | 29.52 |
| 2D-TAN | FS | 39.81 | 23.25 | 79.33 | 52.15 |
| RaNet | FS | 43.87 | 26.83 | 86.67 | 54.22 |
| DRN | FS | 45.40 | 26.40 | 88.01 | 55.38 |
| MMN | FS | 47.31 | 27.28 | 83.74 | 58.41 |
| G2L | FS | 47.91 | 28.42 | 84.80 | 59.33 |
| MomentDiff | FS | 53.79 | 30.18 | - | - |
| IVG-DCL | FS | 50.24 | 32.88 | - | - |
| SCN | WS | 23.58 | 9.97 | 71.80 | 38.87 |
| CTF | WS | 27.30 | 12.90 | - | - |
| WSTAN | WS | 29.35 | 12.28 | 76.13 | 41.53 |
| ICVC | WS | 31.02 | 16.53 | 77.53 | 41.91 |
| MARN | WS | 31.94 | 14.18 | 70.00 | 37.40 |
| CCL | WS | 33.21 | 15.68 | 73.50 | 41.87 |
| CRM | WS | 34.76 | 16.37 | - | - |
| CNM | WS | 35.15 | 14.95 | - | - |
| VCA | WS | 38.13 | 19.57 | 78.75 | 37.75 |
| LCNet | WS | 39.19 | 18.17 | 80.56 | 45.24 |
| **G2L+Ours** | **-** | **55.77** | **32.97** | **91.38** | **60.39** |

Table 15: Video-to-text comparisons on MSRVTT without text augmentation.

| Method | R@1 ↑ | R@5 ↑ | R@10 ↑ | MdR ↓ | MnR ↓ |
|--------|-------|-------|--------|-------|-------|
| *CLIP-ViT-B/32* | | | | | |
| CLIP4Clip (Luo et al., 2022) | 42.7 | 70.9 | 80.6 | 2.0 | 11.6 |
| **+Ours** | **44.2** | **73.8** | **84.3** | **2.0** | **10.4** |
| CenterCLIP (Zhao et al., 2022) | 42.8 | 71.7 | 82.2 | 2.0 | 10.9 |
| **+Ours** | **44.5** | **73.0** | **84.1** | **1.0** | **9.7** |
| X-Pool (Gorti et al., 2022) | 44.4 | 73.3 | 84.0 | 2.0 | 9.0 |
| **+Ours** | **45.8** | **76.4** | **87.3** | **1.0** | **7.5** |
| TS2-Net (Liu et al., 2022) | 45.3 | 74.1 | 83.7 | 2.0 | 9.2 |
| **+Ours** | **48.6** | **77.5** | **85.7** | **2.0** | **7.9** |
| DiffusionRet (Jin et al., 2023) | 47.7 | 73.8 | 84.5 | 2.0 | 8.8 |
| **+Ours** | **51.0** | **75.9** | **87.4** | **2.0** | **6.9** |
| UATVR (Fang et al., 2023) | 46.9 | 73.8 | 83.8 | 2.0 | 8.6 |
| **+Ours** | **49.7** | **75.6** | **86.4** | **1.0** | **7.3** |
| T-MASS (Wang et al., 2024) | 47.7 | 78.0 | 86.3 | 2.0 | 8.0 |
| **+Ours** | **51.5** | **79.9** | **89.8** | **1.0** | **6.4** |
| *CLIP-ViT-B/16* | | | | | |
| X-Pool (Gorti et al., 2022) | 46.4 | 73.9 | 84.1 | 2.0 | 8.4 |
| **+Ours** | **50.2** | **77.4** | **86.3** | **2.0** | **6.1** |
| TS2-Net (Liu et al., 2022) | 46.6 | 75.9 | 84.9 | 2.0 | 8.9 |
| **+Ours** | **48.8** | **78.3** | **86.1** | **1.0** | **7.6** |
| CenterCLIP (Zhao et al., 2022) | 47.7 | 75.0 | 83.3 | 2.0 | 10.2 |
| **+Ours** | **49.8** | **78.0** | **86.4** | **2.0** | **6.5** |
| UATVR (Fang et al., 2023) | 48.1 | 76.3 | 85.4 | 2.0 | 8.0 |
| **+Ours** | **50.9** | **77.4** | **90.5** | **2.0** | **6.8** |
| T-MASS (Wang et al., 2024) | 50.9 | 80.2 | 88.0 | 1.0 | 7.4 |
| **+Ours** | **53.7** | **84.2** | **91.5** | **1.0** | **3.4** |

Table 16: Ablation study on different word types on the VSG task.

| Model | ActivityNet Captions | | | |
|---|---|---|---|---|
| | R@1 IoU=0.3 | R@1 IoU=0.5 | R@5 IoU=0.3 | R@5 IoU=0.5 |
| w/o Noun | 58.87 | 44.23 | 83.90 | 78.04 |
| w/o Verb | 59.15 | 44.92 | 83.71 | 78.95 |
| w/o Subject | 59.36 | 45.78 | 84.02 | 79.88 |
| **Full** | **60.90** | **46.86** | **84.39** | **80.62** |
| Model | Charades-STA | | | |
| | R@1 IoU=0.5 | R@1 IoU=0.7 | R@5 IoU=0.5 | R@5 IoU=0.7 |
| w/o Noun | 31.14 | 25.88 | 74.29 | 45.28 |
| w/o Verb | 31.70 | 26.25 | 73.98 | 45.33 |
| w/o Subject | 33.64 | 27.61 | 74.13 | 46.45 |
| **Full** | **34.85** | **27.96** | **74.28** | **46.70** |

Table 17: Effectiveness comparison with text augmentation under official train/test splits, where "FS" denotes "fully-supervised" and "WS" means "weakly-supervised".

| Method | Type | ActivityNet Captions | | | | Charades-STA | | | |
|---|---|---|---|---|---|---|---|---|---|
| | | R@1, IoU=0.3 | R@1, IoU=0.5 | R@5, IoU=0.3 | R@5, IoU=0.5 | R@1, IoU=0.3 | R@1, IoU=0.5 | R@5, IoU=0.3 | R@5, IoU=0.5 |
| CTRL | FS | 20.27 | 19.40 | 47.82 | 40.78 | 10.32 | 3.54 | 36.98 | 13.40 |
| **+Ours** | **FS** | **22.85** | **21.73** | **48.66** | **43.12** | **12.88** | **4.19** | **37.46** | **16.72** |
| 2D-TAN | FS | 48.32 | 29.38 | 71.36 | 62.30 | 20.18 | 11.35 | 47.05 | 33.82 |
| **+Ours** | **FS** | **51.86** | **32.64** | **72.98** | **63.75** | **23.99** | **14.75** | **49.22** | **34.18** |
| DRN | FS | 48.94 | 30.26 | 69.34 | 64.79 | 23.51 | 13.76 | 47.35 | 34.10 |
| **+Ours** | **FS** | **50.75** | **32.92** | **73.86** | **67.48** | **26.44** | **15.83** | **49.07** | **36.52** |
| RaNet | FS | 52.93 | 31.42 | 73.80 | 65.73 | 22.86 | 15.73 | 46.21 | 30.49 |
| **+Ours** | **FS** | **53.88** | **33.74** | **75.96** | **68.31** | **23.11** | **16.97** | **48.05** | **33.21** |
| MMN | FS | 55.30 | 31.76 | 74.88 | 71.52 | 25.33 | 18.80 | 45.97 | 35.08 |
| **+Ours** | **FS** | **58.76** | **33.08** | **75.33** | **73.59** | **26.87** | **22.48** | **46.03** | **37.85** |
| G2L | FS | 55.75 | 33.01 | 75.25 | 70.89 | 26.54 | 19.85 | 48.06 | 36.70 |
| **+Ours** | **FS** | **60.90** | **46.86** | **84.39** | **80.62** | **34.85** | **27.96** | **74.28** | **46.70** |
| ICVC | WS | 21.88 | 18.59 | 42.76 | 36.82 | **9.27** | **6.89** | 37.55 | 13.98 |
| **+Ours** | **WS** | **22.36** | **20.17** | **44.19** | **38.77** | 8.64 | **6.89** | **39.57** | **16.63** |
| LCNet | WS | 30.15 | **22.08** | 45.80 | 39.25 | 20.48 | 16.61 | 42.32 | 20.89 |
| **+Ours** | **WS** | **33.94** | 21.73 | **46.35** | **40.57** | **23.72** | **17.59** | **43.80** | **23.77** |
| VCA | WS | 31.74 | 25.37 | 46.98 | 42.76 | 17.87 | 12.39 | 45.70 | 22.13 |
| **+Ours** | **WS** | **32.99** | **28.56** | **48.31** | **44.07** | **18.63** | **15.72** | **46.17** | **23.88** |
| WSTAN | WS | 33.72 | 25.74 | 49.30 | 45.88 | 8.15 | 5.43 | 35.27 | 11.86 |
| **+Ours** | **WS** | **35.20** | **27.99** | **51.84** | **48.69** | **10.77** | **6.92** | **37.40** | **13.88** |
| CNM | WS | 35.72 | 28.95 | 50.06 | 48.72 | 14.34 | 9.65 | 43.88 | 18.79 |
| **+Ours** | **WS** | **39.56** | **31.77** | **52.88** | **51.99** | **16.83** | **12.05** | **45.60** | **21.64** |

Table 18: Ablation study on different augmentation approaches.

| | ActivityNet Captions | | | |
|---|---|---|---|---|
| Model | R@1 IoU=0.3 | R@1 IoU=0.5 | R@5 IoU=0.3 | R@5 IoU=0.5 |
| w/o ChatGPT | 58.73 | 44.20 | 82.19 | 78.55 |
| w/o Bard | 59.67 | 44.95 | 83.03 | 79.36 |
| w/o LLaMA | 59.88 | 45.40 | 83.25 | 79.57 |
| **Full** | **60.90** | **46.86** | **84.39** | **80.62** |
| | Charades-STA | | | |
| Model | R@1 IoU=0.5 | R@1 IoU=0.7 | R@5 IoU=0.5 | R@5 IoU=0.7 |
| w/o ChatGPT | 33.04 | 26.93 | 73.50 | 45.27 |
| w/o Bard | 33.92 | 27.40 | 73.83 | 45.62 |
| w/o LLaMA | 34.08 | 27.46 | 73.55 | 45.39 |
| **Full** | **34.85** | **27.96** | **74.28** | **46.70** |

Table 19: Ablation study on for the negative samples on the NExT-QA VideoQA dataset with BLIP-2 as our base model.

| Model | Temporal↑ | Causal↑ | Description↑ |
|---|---|---|---|
| w/o negative samples | 61.3 | 63.4 | 71.5 |
| **w/ negative samples** | **69.2** | **70.1** | **78.4** |

Table 20: Ablation study on for the negative samples on the NExT-QA VideoQA dataset with BLIP-2 as our base model.

| Model | Temporal↑ | Causal↑ | Description↑ |
|---|---|---|---|
| w/o attribute | 60.9 | 62.5 | 67.0 |
| **w/ attribute** | **69.2** | **70.1** | **78.4** |

Table 21: Ablation study on for the negative samples on the NExT-QA VideoQA dataset with BLIP-2 as our base model.

| Model | Temporal↑ | Causal↑ | Description↑ |
|---|---|---|---|
| w/o corse-grained language alignment | 64.3 | 67.5 | 72.3 |
| **w/ corse-grained language alignment** | **69.2** | **70.1** | **78.4** |

Table 22: Ablation study on for the negative samples on the NExT-QA VideoQA dataset with BLIP-2 as our base model.

| Model | Temporal↑ | Causal↑ | Description↑ |
|---|---|---|---|
| w/o fine-grained language alignment | 65.7 | 66.2 | 68.8 |
| **w/ fine-grained language alignment** | **69.2** | **70.1** | **78.4** |

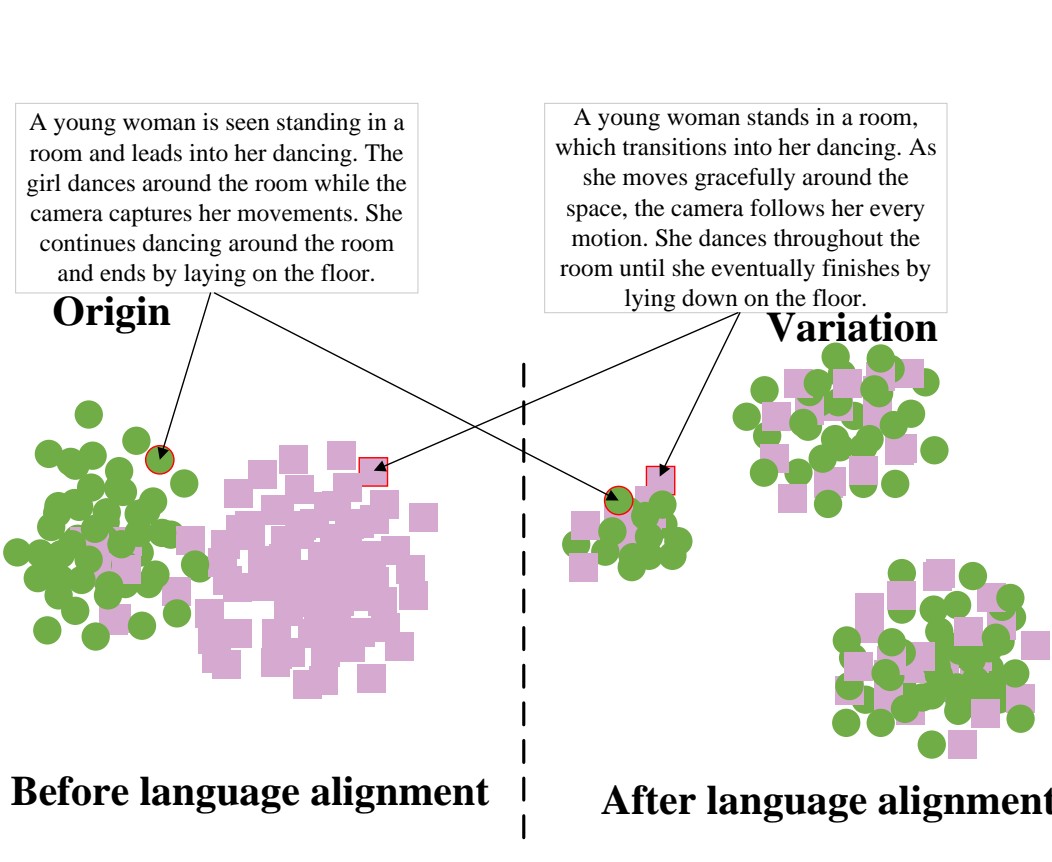

Figure 7: The t-SNE visualizations of "before language alignment" and "after language alignment". Green circles denote the original sentences, while purple triangles denote denote the augmented sentences.

