# OpenReview forum: "Does Your Video-language Model Actually Understand the Language Input?"
_ICLR.cc/2025/Conference — ICLR 2025 Conference Withdrawn Submission_

### Official Review · Reviewer_6Arw · 2024-11-01

**Soundness:** 3
**Presentation:** 3
**Contribution:** 2
**Rating:** 5
**Confidence:** 5

**Summary:**

This paper introduces a text-augmented approach to enhance Video-Language Models (VLMs), addressing the limitations of strictly predefined text inputs in real-world applications. By generating diverse text samples and employing multi-level contrastive learning, the method captures both coarse- and fine-grained semantics for improved video-text alignment. Extensive experiments demonstrate that the proposed framework acts as a plug-and-play module, enhancing the performance of state-of-the-art VLMs across various video-language tasks.

**Strengths:**

1. The proposed method improves VLMs by allowing flexible, user-friendly text inputs rather than rigid predefined templates, making it more practical for real-world applications.
2. The framework is designed as a plug-and-play module, enabling easy integration into existing state-of-the-art VLMs and achieving notable performance boosts across diverse video-language tasks.
3. The organization of this paper is logical.

**Weaknesses:**

1. The text rewriting technique emphasized in this paper is widely used in many tasks, and using LLMs to modify text as a form of data augmentation seems overly simplistic. This approach has already become common knowledge across various research fields, which may weaken the novelty and contribution of the proposed method.
2. In Figure 1, simply presenting a few negative examples to suggest that previous methods cannot leverage their weak text encoders to learn discriminative textual representations lacks persuasiveness. It would be more convincing if the authors provided statistical data, such as the relationship between text representations and model output logits, to better validate their stated motivation.
3. Regarding Q1, one of my main concerns is that the proposed method appears to be a combination of existing solutions, _i.e._, LLM-based text rewriting and contrastive learning with positive and negative samples. Could the authors further discuss the innovation of their proposed approach?
4. Other issues:

(1) Line 93, '_we generate positive samples that lie relatively far from the anchor in the embedding space..._', Given that only certain attributes of the sentences were modified, it is unclear whether they truly diverge in the embedding space. It would be helpful to provide a distribution plot based on t-SNE to visualize this.

(2) Figures deserve further refinement, _e.g._, Fig. 4 (3) contains unnecessary coordinate points.

**Questions:**

Please focus on Weaknesses 1 and 3, and I encourage the authors to provide further discussion and clarification on these points.

---

> ### Author Response · Authors · 2024-12-03
>
> **About novelty.**
>
> Sorry for your misunderstanding on the novelty. 1) About LLM. We do not directly use LLM for data annotation. We only borrow knowledge from LLM. In our paper, we generate multi-level texts to fully understand the given text for coarse-grained language alignment, different from previous single-level text generation methods. Many LLM-based methods only treat each word with the same significance, which leads to that some unimportant words (e.g., article, particle) have a large weight. In realistic language rewriting, some core words (e.g., noun) tend to stay the same. Different from these methods, we can evaluate the significance score of each word to better understand the text.
> Also, we generate attributes to reason the fine-grained semantics in the given sentence and utilize attribute sampling to purify these semantics-rich attributes, which is a significant difference between our framework and other LLM-based methods. Moreover, these semantics-rich attributes can be used for cross-modal matching. For example, "person" in text always corresponds to a person with a head, two eyes and two arms.
> 2) Different from previous representation learning-based methods that only construct positives and negatives from a level, we construct positives and negatives from different levels (word-level and structure-level). Besides, we design a brand-new weighted sentence incorporation module for cross-modal fusion. Moreover, our attribute-based text reasoning can effectively mine the core semantics by semantic-based selection and format-based selection, which will reduce the negative impact of noise from generated texts.
>
> **Other comments**
>
> About the relationship between text representations and model output. We use t-SNE to visualize the feature in Figure 6 in the appendix.
>
> We use t-SNE to visualize the feature in Figure 6 in the appendix. In the figure, without our language alignment modules (coarse-grained language alignment and fine-grained language alignment), the feature distance of two semantics-similar sentences is very far in the latent space. It shows that only utilizing previous text encoders cannot effectively map these semantics-similar sentences into a feature cluster. With our language alignment modules, the features of the two sentences are very close.

---

### Official Review · Reviewer_fj8w · 2024-11-03

**Soundness:** 3
**Presentation:** 3
**Contribution:** 2
**Rating:** 6
**Confidence:** 3

**Summary:**

Vision-language models typically depend on massive datasets of image/video-text pairs. However, these text descriptions are often noisy and abbreviated, limiting the model’s performance. This work addresses these challenges by enhancing the text encoder's ability to produce consistent representations for semantically similar texts in real-world multi-modal datasets. Additionally, they integrate generated samples as supplementary data alongside the original training samples, utilizing a novel weighted contrastive loss to optimize their impact and ensure a balanced contribution to training.

**Strengths:**

1. This work conducts extensive experiments on popular architectures, including All-In-One and Just Ask, demonstrating substantial performance improvements.

2. Enhancing text diversity and knowledge is a valuable approach that has been partially overlooked in previous research, yet it proves highly effective in this study.

3. Beyond word-level and sentence-level augmentations, this work also introduces structure-level augmentation to further enrich the model's understanding.

**Weaknesses:**

1. This method leverages multiple off-the-shelf large language models to generate refined text representations. Bring additional training cost and data scale.

2. The dataset contains noisy and incorrect annotations, which may increase challenges by potentially amplifying misalignment when relying on LLMs to hypothesize connections.

3. A similar concept is utilized in retrieval-augmented methods, which also aim to enhance data quality by incorporating diverse sources.

**Questions:**

1. In Table5, if the running time include time consuming over text repharase by LLM? If not, can you include the baseline running time in this table also?

---

> ### Author Response · Authors · 2024-12-03
>
> *Q1: This method leverages multiple off-the-shelf large language models to generate refined text representations. Bring additional training cost and data scale.*
>
> About training cost and data scale. We add the time consuming over text repharased by LLM in Table 5. Although we generate some texts, three significant strategies (attribute sampling, semantic-based selection and format-based selection) are utilized to select some texts/attributes for cross-modal fusion, which can effectively reduce the computational cost.
>
> *Q2: The dataset contains noisy and incorrect annotations, which may increase challenges by potentially amplifying misalignment when relying on LLMs to hypothesize connections.*
>
> In fact, we notice some noisy and incorrect annotations in the text input. Fortunately, we can utilize attribute generation and attribute sampling to mine the ground-truth text semantics. Also, our semantic-based selection and format-based selection can select the correct attributes for fine-grained language alignment.
>
> *Q3: A similar concept is utilized in retrieval-augmented methods, which also aim to enhance data quality by incorporating diverse sources.*
>
> Our main aim is that most current VLMs implicitly assume that all the texts are predefined by the specific template. These retrieval-augmented methods directly enhance data quality by incorporating diverse sources, which increases the computational cost. Unlike them, we only choose some texts/attributes for cross-modal fusion. Also, these methods cannot remove the noisy and incorrect annotations, which might limit their performance. Different from them, we design three significant strategies (attribute sampling, semantic-based selection and format-based selection) to only mine some significant language knowledge for cross-modal fusion, which can reduce the negative impact of noises.

---

### Official Review · Reviewer_9qRs · 2024-11-10

**Soundness:** 1
**Presentation:** 2
**Contribution:** 2
**Rating:** 5
**Confidence:** 5

**Summary:**

This paper proposes a text-augmented Video-Language Model (VLM) method to improve video-text fusion. The key idea is to generate diverse sentences from the original ones using a pre-trained LLM, and then apply a multi-level contrastive learning module to mine both coarse-grained and fine-grained textual semantics. This approach is shown to effectively improve the performance of state-of-the-art VLM models on various video-language tasks.

**Strengths:**

1. The proposed method is intuitive.
2. The paper is easy to follow.
3. The current empirical results look strong.

**Weaknesses:**

1. The current review of existing work on video-language model is severely incomplete, which makes the motivation that "video language models don't understand language input" actually invalid. There are many existing methods [a,b,c,d,e] relying on text-based representation for video input and they have already shown great effectiveness in video language modeling, with strong pretrained LLM as the initialization. It is important for the current work to comprehensively review the progress in the community and position more clearly.

2. It is also important to compare with at least some of the existing methods. Currently the authors adopts a rather implicit approach of doing text augmentation. However, the existing methods [a,b,c,d,e] directly take text representation as input which shows great generalization and can even work in training-free setting [b,d,e].

3. Empirically, the motivation is not really validated in the current empirical results. It is not clear whether the proposed method improves the performance because it is more invariant to the text prompts or actually because the video information is better utilized through the decomposition to attributes. It is important to show more analysis on why the proposed method works.
- Check the dynamics of the model during training time in terms of the performance on training data and evaluation data.
- Check the cross-data evaluation performance.


a. VX2TEXT: End-to-End Learning of Video-Based Text Generation From Multimodal Inputs, CVPR 2021

b. Language Models with Image Descriptors are Strong Few-Shot Video-Language Learners, Neurips 2022

c. Towards Fast Adaptation of Pretrained Contrastive Models for Multi-channel Video-Language Retrieval, CVPR 2023

d. Socratic Models: Composing Zero-Shot Multimodal Reasoning with Language, 2022

e. Training-free Deep Concept Injection Enables Language Models for Video Question Answering, 2023

**Questions:**

Please check weakness for details.

---

> ### Author Response · Authors · 2024-12-03
>
> *Q1: The current review of existing work on video-language model is severely incomplete, which makes the motivation that "video language models don't understand language input" actually invalid.*
>
>  [a] proposes a framework called VX2TEXT, which is used to generate text from multimodal inputs containing video, text, speech or audio. The main objectives of VX2TEXT are: 1) to extract meaningful information from each input modality; 2) to effectively fuse information from different modalities; 3) to generate human-understandable text output.
> [b] presents a new method called LaViLa, which uses large language models (LLMs) to learn video-language representations. [b] fine-tunes pre-trained LLMs to be conditioned on visual input and further fine-tuning them to create automatic video narration.
> [c] proposes a method for fast adaptation of pre-trained contrastive models for multi-channel video-language retrieval. The authors explored two dimensions: video representation (continuous feature vectors or discrete text tokens) and video-text fusion method (multimodal Transformer or pre-trained contrastive text model).
> [d] shows that this diversity is symbiotic, and can be leveraged through Socratic Models (SMs): a modular framework in which multiple pretrained models may be composed zero-shot i.e., via multimodalinformed prompting, to exchange information with each other and capture new multimodal capabilities, without requiring finetuning.
> [e] propose a novel approach to enabling pretrained language models to perform video question answering without any training. The proposed Deep Concept Injection effectively circumvents the necessity of training projection networks, a widely accepted practice in this field, and instead makes insightful use of observed visual concepts as additional input text tokens and as a means for augmenting intermediate features.
>
> Different from them, we propose a novel text-augmented VLM method to improve video-text fusion by text rewriting. Specifically, we first generate various text samples from the original ones based on the pre-trained LLM to target specific text components. A multi-level contrastive learning module is designed to mine the coarse-grained language information. Moreover, we also propose an attribute-based text reasoning strategy to learn fine-grained textual semantics.
>
> In the final version, we will add a comprehensive review about video-language models.
>
> *Q2: It is also important to compare with at least some of the existing methods.*
>
> In fact, the text augmentation method is not our main contribution. Our main contribution is to propose a simple yet highly effective framework to improve the robustness and performance of VLMs, which can serve as a plug-and-play module for state-of-the-art VLMs. We try to reduce the feature distance between semantics-similar sentences. Due to the page limitation and the short author-reviewer discussion, we will add the comparison with [a-e] in the final version. In the future, we will introduce more state-of-the-art works (e.g., wikiHow and TTC-Loc) into our framework for better performance.
>
> Moreover, our framework can be used in the training-free setting as follows:
>
> **VSG task with "R@1, IoU=0.3" as evaluation metric**
>
> Method  | Charades-STA | ActivityNet Captions|
> -------------------|------------------|-------------------
> TFVTG [3] (without text aug) | 67.04 | 49.34 |
> +Ours | 68.15 | 50.86 |
> TFVTG [3] (with text aug) | 58.11 | 37.52 |
> +Ours |  63.20 | 45.27 |
>
> [3] Training-free Video Temporal Grounding using Large-scale Pre-trained Models, ECCV 2024
>
> *Q3: Empirically, the motivation is not really validated in the current empirical results.*
>
> In fact, Figure 4 shows the dynamics of the model during training time in terms of the performance, which illustrates the effectiveness of each module.
>
> **Cross-data evaluation for zero-shot VideoQA task with WebVid10M as training data**
>
> Method  | iVQA | MSRVTT-QA | MSVD-QA | ActivityNet-QA | TGIF-QA|
> -------------------|------------------|-------------------|------------------|-------------------|-------------------
> FrozenBiLM [4] (without text aug) | 26.2| 16.9| 33.7| 25.9| 41.9|
> +Ours | 28.1| 19.5| 36.0| 26.4| 44.1|
> FrozenBiLM [4] (with text aug) | 17.3| 12.8| 25.7| 20.9| 33.4|
> +Ours | 24.6| 15.7| 29.0| 24.3| 38.0|
>
>
> [4] Zero-Shot Video Question Answering via Frozen Bidirectional Language Models, NeurIPS 2022
>
> Also, we will add more analysis and experiments in the final version.

---

### Official Review · Reviewer_xpsD · 2024-11-11

**Soundness:** 3
**Presentation:** 2
**Contribution:** 2
**Rating:** 5
**Confidence:** 3

**Summary:**

This paper aims to close the video-text alignment gap using data augmentation techniques particularly designed for textual prompt input.

The proposed method involves prompting LLMs to modify the anchor text to derive positive/negative text that is semantically simmilar/disimilar to the original one. The modification is done on both word-level and structure-level, and there is an attribute-level selection module deisgned to guarantees the quality of the generated textual labels.

Given a triplet of (positive text, negative text, video embedding), the contrastive loss is calculated and minimized so that the cosine similarity between the visual and the positive/negative is maximized/minimized. In each sentence, the word with the highest constrastive loss is considered.

The authors validated the effectiveness of their augmentation technique extensively on three downstream tasks: 1) video sentence grounding, 2) video question answering, 3) video-text retrieval. The experiment results show enhanced performance on almost all tasks when paired with all relevent methods.

**Strengths:**

1. The paper is well-motivated. The significant issue of multi-modality misalignment between video and text has not been widely investigated.
2. The proposed method of using a contrastive loss to encourage VLM to distinguish between positively and negatively augmented textual prompts is sound and intuitive. The plug-in nature of the textual augmentation technique makes it applicable to all relevant methods in the area of VLM.
3. A significant amount of effort in this work is devoted to experiments, and the reported empirical results strongly back up the effectiveness of the method on all three tasks when paired with all VLM methods.
4. The ablation study offers a comprehensive look at the importance of each module. The analysis of the word type is quite insightful.

**Weaknesses:**

1. The approach lacks novelty. The idea of  1) using LLMs for data annotation and augmentation and 2) constrasting positives and negatives for representation learning has been well studied.
2. Based on the context of video-text modality alignment, there is a need to have a discussion comparing it to the well-studied image-text setting. How is the misalignment in the image-text domain different from the video-text one? What particular considerations have been taken for it?
3. Further clarafication is needed for the methodology. Are both video encoder and textual encoder being updated? There is no gradient backpropagation indicated in Figure 2.
4. The approach to only minimize the constrastive loss on "the most discriminative word" is not properly justified. And the loss is definitely not "Weighted contrastive loss" as indicated in equation (4).
5. The ablation study on hyperparameters is missing.

**Questions:**

Please refer to the weaknesses above.

---

> ### Author Response · Authors · 2024-12-03
>
> *Q1: The idea of 1) using LLMs for data annotation and augmentation and 2) constrasting positives and negatives for representation learning has been well studied.*
>
> **About novelty**. Sorry for your misunderstanding on the novelty. 1) About LLM. We do not directly use LLM for data annotation. We only borrow knowledge from LLM. In our paper, we generate multi-level texts to fully understand the given text for coarse-grained language alignment, different from previous single-level text generation methods. Many LLM-based methods only treat each word with the same significance, which means that some unimportant words have a large weight. We can evaluate the significance score of each word to better understand the text.
> Also, we generate attributes to reason the fine-grained semantics in the given sentence and utilize attribute sampling to purify these semantics-rich attributes, which is a significant difference between our framework and other LLM-based methods.
> 2) Different from previous representation learning-based methods that only construct positives and negatives from a level, we construct positives and negatives from different levels (word-level and structure-level). Besides, we design a brand-new weighted sentence incorporation module for cross-modal fusion. Moreover, our attribute-based text reasoning can effectively mine the core semantics by semantic-based selection and format-based selection, which will reduce the negative impact of noise from generated texts.
>
> *Q2: Based on the context of video-text modality alignment, there is a need to have a discussion comparing it to the well-studied image-text setting. How is the misalignment in the image-text domain different from the video-text one? What particular considerations have been taken for it?*
>
> **Video-text modality alignment vs image-text modality alignment**. Image only contains the appearance information (spatial information), but video contains much temporal information. Most video-text datasets focus on the video actions, and they pay more attention to the temporal information. Besides, their sentences are also different, i.e., video-text datasets contain much action information with various verbs. For the image-text datasets, the text does not contain verbs since there is no action information.
>
> *Q3: Further clarafication is needed for the methodology. Are both video encoder and textual encoder being updated? There is no gradient backpropagation indicated in Figure 2.*
>
> **About encoders**. Since our framework is plug-and-play, both video encoders and textual encoders follow the original baseline models. Most of these baseline methods use pre-trained encoders without updates.
>
> *Q4: The approach to only minimize the constrastive loss on "the most discriminative word" is not properly justified. And the loss is definitely not "Weighted contrastive loss" as indicated in equation (4).*
>
> **About constrastive loss**. Sorry for the misunderstanding. We accidentally deleted some texts and the weighted constrastive loss in our first version. Now, the weighted constrastive loss is available in Eq. (5).
>
> *Q5: The ablation study on hyperparameters is missing.*
>
> **About the ablation study on hyperparameters**. We have reported it in Figure 6 in the appendix.

---

### Official Review · Reviewer_CeGt · 2024-11-11

**Soundness:** 2
**Presentation:** 2
**Contribution:** 2
**Rating:** 5
**Confidence:** 4

**Summary:**

This paper focuses on the settings that, current LLMs rely on predefined text templates. The authors propose a new text-augmented VLLM method that uses text rewriting to improve video-text fusion. The method has three main parts:

* Text sample generation with pre-trained LLMs
* Multi-level contrastive learning for coarse-grained language processing
* Attribute-based text reasoning for fine-grained semantic understanding

The proposed approach can be actively integrated with existing VLLMs as a plug-and-play module, and experimental results demonstrate improved performance across multiple vision-and-language tasks.

**Strengths:**

* The authors carefully design a LLM-based text augumentation methods for video-language tasks. The language alignment following the augumentation is convicing.
* The authors conduct intensive experiments with many VLLMs across various benchamarks to prove the effectiveness of the method.

**Weaknesses:**

* The storyline and frame of work seems somehow not that well-aligned. The authors poses the paper as "Does Your Video-language Model Actually Understand the Language Input?", but the analysis and findings don't fully address this core question, like in what aspects, the current VLM fails to understand the language inputs, and how the proposed method specifically addresses these understanding gaps. Some claims also need further justification, such as "predefined text inputs are too strict and user-unfriendly". For example, in video captioning tasks, I don't think there are some predefined template things, the labeler usually write natural captions just following some guidelines.

* Mostly, the paper compare the different components they proposed in their own methods. But there are lots of previous non-LLM data augmentation methods in NLP [1]. Some conventional methods, e.g., synonym/antonym replacement, back-translation, paraphrasing, may work as well. These baselines could provide valuable context for evaluating the method's effectiveness.

* The choice of benchmarks could be strengthened. For instance, in video captioning tasks, I think currently ActyNet-Caps, Charades should not be very comprehensive and convincing benchmarks. Some recent benchmark like VATEX [3] (tranditional video captioning, more longer and complex captions, multiple references for each video instead of 1) or YouCook2 [4] (dense video captioning, also with complex captions) could be better choices. There are also some recent benchmarks, specifically designed for evaluating Video-LLMs, are also missing (like Video-MME).

* This raises me another questions that I saw most of text cases in these paper are quite short, like in just a few words. I am not sure whether this method is scalable and extendable to longer and complex real-world texts. It would be better that authors give some more complex cases, and show the distribution on length of their augumented texts.


References:
[1] Data augmentation approaches in natural language processing: A survey, Li et al, AI Open 2022.

[2] VATEX: A Large-Scale, High-Quality Multilingual Dataset for Video-and-Language Research, Wang et al, ICCV 2019.

[3] Towards Automatic Learning of Procedures from Web Instructional Videos, Zhou et al, AAAI 2018.

[4] Video-MME: The First-Ever Comprehensive Evaluation Benchmark of Multi-modal LLMs in Video Analysis. Fu et al.

**Questions:**

Most of my questions and concerns are already listed in the Weakness section, here I leave some other questions:

* Some implementation details are not quite clear to me. The proposed methods can be integrated with any VLLMs, and for the main experiments, on which datasets the authors use their methods? For instance, for the ActivityNet Captions task, does the authors use the training dataset to get more augumented cases and fine-tune the open-source VLLMs? Please correct me if I misunderstood.

* The proposed methods can generate both positive and negative augumented texts for videos. It would be better to have the ablation study on the effectiveness of using negative texts, especially for tasks like captioning/generation.

---

> ### Author Response · Authors · 2024-12-03
>
> *Q1: The storyline and frame of work seems somehow not that well-aligned. *
>
> **About the storyline and frame of work**. In the revision version, we have stated the answer in the conclusion section: “Most VLMs cannot fully understand the text input, but we can conduct a simple and plug-and-play module to help them understand the text input.” We observe that overly strict text input is user-unfriendly in some multi-modal tasks, which will limit the model performance. Due to the page limitation, we only choose three representative tasks (VSG, VideoQA, VTR). As for the video captioning task, different labelers have various language usage habits. For example, for a video on playing tennis, a labeler writes a natural caption “two persons play tennis”, while another labeler might give a detailed caption “A lady in a red T-shirt and a man in blue play tennis in the playground.” With various captions for training, we will obtain different results. Fortunately, in Section 3.2, we can generate attributes for mining the latent language semantics of the two captions for fine-grained language alignment.
>
> *Q2: Some conventional methods, e.g., synonym/antonym replacement, back-translation, paraphrasing, may work as well.*
>
> **About different non-LLM data augmentation methods**. We compare these methods with our methods on the NExT-QA VideoQA dataset with BLIP2 as our base model as follows:
>
> Method  | Temporal | Causal  | Description |
> -------------------|------------------|-------------------|------------------
> synonym replacement |	64.4	| 62.3 |	71.0 |
> back-translation |	62.8 |	62.7 |	68.9|
> paraphrasing|	65.3|	63.4|	72.5|
> Ours|	69.2|	70.1|	78.4|
>
> *Q3: The choice of benchmarks could be strengthened.*
>
> Since many multi-modal works use traditional datasets for performance evaluation, we follow their setting for fair comparison. Also, our framework can work on other large-scale datasets as follows:
>
> **Text-to-video retrieval task on VATEX**
>
> Model|	R@1|	R@5|	R@10|	MedR|
> -------------------|------------------|-------------------|------------------|------------------
> HGR [5] (without text augmenation)|	35.1|	73.5|	83.5|	2|
> +Ours|	39.4|	76.1|	86.3|	1|
> HGR [5] (with text augmenation)|	26.8|	62.8|	74.9|	4|
> +Ours|	32.7|	69.5|	78.8|	2|
>
> **Text-to-video retrieval task on YouCook2**
>
> Model|	R@1|	R@5|	R@10|	MedR|
> -------------------|------------------|-------------------|------------------|------------------
> StarVR [6](without text augmenation)|	13.1|	32.6|	44.1|	16|
> +Ours|	16.4|	35.8|	49.2	|4|
> StarVR [6](with text augmenation)|	8.6|	24.7|	32.9|	24|
> +Ours|	11.2|	27.9|	40.8|	18|
>
> **Performance (8 frames) on Video-MME without subtitles**
>
> Model|	Short|	Medium|	Long|	Overall|
> -------------------|------------------|-------------------|------------------|------------------
> LongVA [7] (with text augmenation)|	55.1|	46.3	|42.1|	47.9|
> +Ours|	57.0|	47.5|	42.9	|49.1|
> LongVA [7](with text augmenation)|	51.4|	40.8|	36.7|	42.9|
> +Ours|	53.6|	44.3|	39.2|	45.7|
>
> *Q4: This raises me another questions that I saw most of text cases in these paper are quite short, like in just a few words.*
>
> In fact, there are more long sentences than short ones. To make the text in the figures clear, we choose some short sentences in our submission. An example on long sentence is shown in Figure 7 in the appendix.For the augmented texts of the ActyNet-Caps dataset, the distribution on length of their augmented texts is as follows: 1-word to 10-word (18.2%), 11-word to 20-word (26.9%), 21-word to 30-word (52.6%), 31-word to 40-word (1.4%), longer than 50-word (0.9%).
>
> [5] Fine-grained Video-Text Retrieval with Hierarchical Graph Reasoning, CVPR 2020
> [6] A large cross-modal video retrieval dataset with reading comprehension, PR 2025
> [7] Long Context Transfer from Language to Vision, Arxiv 2024
>
> *Q5: Some implementation details are not quite clear to me. *
>
> For instance, for the ActivityNet Captions dataset, we get more augmented texts on the training set, test set and validation set. By adding corresponding video, we can get the augmented video-text pairs. For the case with text augmentation, we combine original pairs and augmented pairs (original training video-text pairs and augmented training video-text pairs for training; original testing video-text pairs and augmented testing  video-text pairs for testing). For the case without text augmentation, we only use original pairs, i.e., original training video-text pairs for training and original testing video-text pairs for testing.
>
> * It would be better to have the ablation study on the effectiveness of using negative texts, especially for tasks like captioning/generation.*
>
> We have shown the ablation study on the negative texts in Table 19 in appendix.

---

### Official Review · Reviewer_m6Ls · 2024-11-19

**Soundness:** 2
**Presentation:** 4
**Contribution:** 2
**Rating:** 5
**Confidence:** 4

**Summary:**

The authors assert that existing video-language models may not fully comprehend the language input, as evidenced by a significant drop in model performance when input texts are rewritten. However, their experiments suggest that this issue may be more attributable to domain-specific fine-tuning rather than a fundamental limitation of the models.

Regarding the authors' proposed solution, a more straightforward approach might involve leveraging video-based large-language models, which possess stronger domain generalization capabilities.



Overall, based on the current experiments, the paper's contribution appears to be limited to domain-specific small models. Please refer to the questions below for further suggestions.

**Strengths:**

- This paper has a clear motivation.
- The proposed method is significant for domain-specific models.

**Weaknesses:**

The observed phenomenon may be more attributable to domain-specific fine-tuning:
- The degradation is more pronounced in datasets with stronger text patterns. For instance, Charades-STA, constructed using text templates, exhibits the most significant performance degradation. This suggests that the phenomenon might stem from domain-specific patterns within the dataset.
- Models with larger pre-trained text encoders, such as BLIP-2 and InternVideo, experience less performance degradation. This indicates that larger models may have greater generalization capabilities when handling text input.

According to the observation above, the authors' solution is limited to domain-specific small models
- Does this phenomenon diminish in larger models like Videochat [1] or Video-LLaVA [2]? While this is a promising and straightforward solution, it has not been considered as a baseline.
- Does this phenomenon persist in zero-shot settings instead of supervised fine-tuning (SFT)? This is crucial for assessing the impact of the proposed methods.

[1] Videochat: Chat-centric video understanding.
[2] Video-LLaVA: Learning United Visual Representation by Alignment Before Projection.

**Questions:**

If the phenomenon persists in large models or zero-shot settings and the authors' solution remains effective, I will consider raising my score.

---

> ### Author Response · Authors · 2024-12-03
>
> *Q1: The observed phenomenon may be more attributable to domain-specific fine-tuning.*
>
> **About domain-specific small model**: In fact, larger pre-trained text encoders, such as BLIP-2 and InternVideo still experience severe performance degradation. Since their tasks, evaluation metrics and datasets are different (BLIP-2 and InternVideo are VideoQA methods on the NExT-QA dataset, while Charades-STA is a VSG dataset), the specific value of performance degradation should not be used for direct comparison. The Charades-STA dataset already contains various text inputs. When augmenting the text inputs, we will get many text queries with various templates with similar semantics, which limits their performance. For the large models, they have some generalization capabilities when handling text input since they have been pre-trained in various text inputs with various templates. Please note that our framework can serve as the plug-and-play module to help some small models achieve performance that rivals that of larger models.
>
> *Q2: According to the observation above, the authors' solution is limited to domain-specific small models*
>
> **About this phenomenon in larger models**: Yes, the phenomenon diminishes in larger models. We check the performance of Videochat [1] or Video-LLaVA [2], where "aug" denotes "augmentation".
>
> **VideoQA task with "accuracy" as evaluation metric**
>
> Method  | MSVD-QA | MSRVTT-QA | TGIF-QA |
> -------------------|------------------|-------------------|------------------
> VideoChat (without text aug) | 56.3 | 45.0 | 34.4 |
> +Ours | 59.4 | 47.5 | 38.9 |
> VideoChat(with text aug) | 41.2 | 33.9 | 25.6|
> +Ours | 48.8 | 40.5 | 31.4 |
> | | | |
> Video-LLaVA (without text aug) | 70.7 | 59.2 | 70.0 |
> +Ours | 72.2 | 63.7 | 71.6 |
> Video-LLaVA (with text aug) | 65.1 | 51.7 | 64.2 |
> +Ours | 69.4 | 56.2 | 68.1 |
>
> **VSG task with "R@1, IoU=0.3" as evaluation metric**
>
> Method  | Charades-STA | ActivityNet Captions|
> -------------------|------------------|-------------------
> TFVTG [3] (without text aug) | 67.04 | 49.34 |
> +Ours | 68.15 | 50.86 |
> TFVTG [3] (with text aug) | 58.11 | 37.52 |
> +Ours |  63.20 | 45.27 |
>
> **zero-shot VideoQA task with WebVid10M as training data**
>
> Method  | iVQA | MSRVTT-QA | MSVD-QA | ActivityNet-QA | TGIF-QA|
> -------------------|------------------|-------------------|------------------|-------------------|-------------------
> FrozenBiLM [4] (without text aug) | 26.2| 16.9| 33.7| 25.9| 41.9|
> +Ours | 28.1| 19.5| 36.0| 26.4| 44.1|
> FrozenBiLM [4] (with text aug) | 17.3| 12.8| 25.7| 20.9| 33.4|
> +Ours | 24.6| 15.7| 29.0| 24.3| 38.0|
>
> Also, we will add more analysis and experiments (e.g., few-shot setting) in the final version.
>
> [3] Training-free Video Temporal Grounding using Large-scale Pre-trained Models, ECCV 2024
>
> [4] Zero-Shot Video Question Answering via Frozen Bidirectional Language Models, NeurIPS 2022

---

### Note · Authors · 2025-01-23

I have read and agree with the venue's withdrawal policy on behalf of myself and my co-authors.